# MedGMAE: Gaussian Masked Autoencoders for Medical Volumetric Representation Learning

**Xueming Fu**[1,2][*], **Fenghe Tang**[1,2][*], **Rongsheng Wang**[1,2], **Yingtai Li**[1,2], **Lixia Han**[5], **Jian Lu**[6], **Zihang Jiang**[1,2][†] **& S. Kevin Zhou**[1,2,3,4][†]

[1]School of Biomedical Engineering, Division of Life Sciences and Medicine, USTC
[2]MIRACLE Center, Suzhou Institute for Advance Research, USTC
[3]Jiangsu Provincial Key Laboratory of Multimodal Digital Twin Technology
[4]State Key Laboratory of Precision and Intelligent Chemistry, USTC
[5]Nanjing University of Aeronautics and Astronautics
[6]Department of Radiology, Medical School, Zhongda Hospital, Southeast University
{jzh0103, skevinzhou}@ustc.edu.cn

## Abstract

Self-supervised pre-training has emerged as a critical paradigm for learning transferable representations from unlabeled medical volumetric data. Masked autoencoder based methods have garnered significant attention, yet their application to volumetric medical image faces fundamental limitations from the discrete voxel-level reconstruction objective, which neglects comprehensive anatomical structure continuity. To address this challenge, We propose MedGMAE, a novel framework that replaces traditional voxel reconstruction with 3D Gaussian primitives reconstruction as new perspectives on representation learning. Our approach learns to predict complete sets of 3D Gaussian parameters as semantic abstractions to represent the entire 3D volume, from sparse visible image patches. MedGMAE demonstrates dual utility across medical imaging applications. For representation learning, sparse Gaussian prediction produces superior encoder representations that outperform traditional MAE baselines on downstream segmentation, classification, and registration tasks. For volumetric reconstruction, the Gaussian decoder leverages pretrained anatomical priors to accelerate 3D CT volume reconstruction convergence. Extensive experiments across multiple medical imaging datasets demonstrate that our approach achieves superior performance, establishing a new framework for medical image pre-training. The code will be available in https://github.com/windrise/MedGMAE.

## 1 Introduction

Volumetric medical imaging modalities, such as Computed Tomography (CT) and Magnetic Resonance Imaging (MRI), have become indispensable cornerstones of modern clinical practice, providing three-dimensional anatomical information crucial for diagnosis, treatment planning, and prognostic assessment. The advent of deep learning has heralded a new era in the automated analysis of these data, demonstrating unprecedented performance across a spectrum of tasks (Zhou et al., 2023c; Litjens et al., 2017). However, the full potential of these data-hungry models is severely constrained by a fundamental bottleneck: the scarcity of large-scale, expertly annotated datasets (Willemink et al., 2020; Ravì et al., 2016).Recent work explores train-free paradigms that leverage pretrained 2D foundation models to extract semantic information from 3D volumes (An et al., 2025), demonstrating an alternative.

---

[*]Equal contributions
[†]Corresponding authors

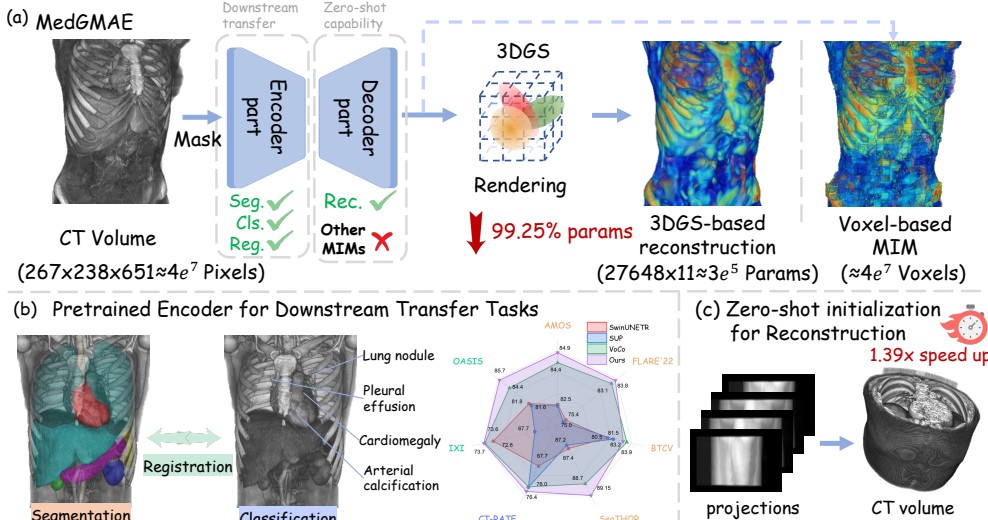

Figure 1: MedGMAE overview. (a) our MedGMAE pre-training with 3D Gaussian Splatting reconstruction leverages CT volume sparsity (anatomical organs occupy only 11.8% of space) to achieve 99.25% parameter reduction and superior coherence compared to voxel-based MIM methods. (b) Pre-trained encoder fine-tuning for downstream tasks: our MedGMAE could learn a strong encoder representation for downstream segmentation, registration, and classification tasks across multiple medical datasets. (c) our MedGMAE could bring a zero-shot capability for 3DGR-based CT reconstruction with 1.39× speed-up.

This challenge sparks an increasing interest in self-supervised pre-training methods that can harness unlabeled 3D data to improve performance in downstream tasks, such as segmentation, registration, and diagnosis. Due to the high anatomical similarity across different medical volumes, Masked Image Modeling (MIM) has emerged as a powerful 3D pre-training approach for learning local representations by reconstructing masked regions from visible context. Despite its promising results, we identify three fundamental yet underexplored challenges that limit the effectiveness of directly reconstructing masked regions via voxel-level regression: **(i) Discrete reconstruction conflicts with anatomical continuity:** conventional MIM methods typically regress discrete intensity voxels of masked regions (He et al., 2022; Chen et al., 2023). While this teaches the model to "fill in blanks" based on immediate spatial context works well for photorealistic data, it is ill-suited for capturing the underlying semantic continuity and geometric abstraction of anatomical structures in volumetric space. Discrete voxel regression often fails to model shape-consistent features, which are crucial for understanding medical images and transferring knowledge to downstream tasks. **(ii) Non-transferable decoder representations:** A common yet overlooked issue in voxel-based MIM is that the decoder is designed purely for reconstructing low-level pixel intensities (Xie et al., 2022b; Tang et al., 2024; Tian et al., 2023). The pre-trained decoder is typically discarded, and the features it learns are rarely leveraged for downstream tasks, while its zero-shot capability is inherently constrained by the reliance on pixel-level reconstruction. **(iii) Sparse anatomical distribution leads to parameter inefficiency:** Unlike natural 2D images that contain dense textural information throughout, 3D medical images are inherently sparse in both semantic and intensity distributions. Redundant voxel-based representation fails to achieve optimal reconstruction efficiency.

To address these limitations, we introduce Medical Gaussian Masked Autoencoder (MedGMAE), a novel self-supervised framework tailored for 3D medical image pretraining grounded in a key insight: *learning sparse 3D Gaussian representations instead of reconstructing dense voxel intensities*. As shown in Fig.1(a), our approach leverages 3D Gaussian primitives as an intermediate representation that naturally addresses the aforementioned challenges through three key advantages: **(i) Continuous geometric modeling for anatomical coherence:** 3D Gaussian primitives provide continuous, differentiable representations that inherently capture geometric abstractions and shape consistency across anatomical structures. Each Gaussian primitive encodes spatial position, orientation, and scale information, enabling the model to learn semantically meaningful geometric features

that align with the continuous nature of anatomical boundaries (as shown in Fig.1(b)). **(ii) Transferable decoder:** Our Gaussian-based decoder remains useful after pre-training, directly serving as sophisticated initialization for Gaussian representation 3D medical reconstruction (as shown in Fig.1(c), faster $1.39\times$ for coverage). **(iii) Parameter-efficient representation:** Our Gaussian-based approach naturally aligns with the sparse anatomical distribution in medical volumes, achieving superior parameter efficiency (99% reduction in parameters).

The main contributions of this work can be summarized as follows:

- First, we introduce MedGMAE, the first framework to successfully adapt and extend Gaussian-based masked autoencoding for self-supervised pre-training on 3D volumetric medical data. Our approach learns parameter-efficient representations that better captures continuous anatomical boundaries, enabling models to develop more structured and anatomically-aware representations.

- Second, we demonstrate a novel application for the pre-trained decoder by using it as a zero-shot, geometry-aware initializer for downstream 3D CT reconstruction tasks. The learned anatomical priors from pre-training significantly accelerate 3D Gaussian Representation-based CT reconstruction convergence, thus bridging self-supervised pre-training with practical medical image reconstruction applications.

- Third, extensive experiments across downstream tasks including segmentation, classification, and registration validate the superiority of our proposed approach compared to voxel-based masked representation methods. Additionally, experiments on low-dose CT reconstruction tasks demonstrate the zero-shot initialization capability of our proposed framework, showing significant acceleration in convergence while maintaining reconstruction quality.

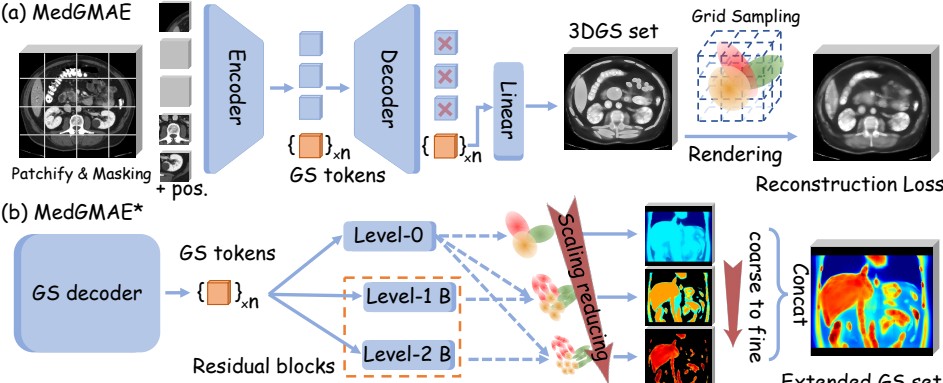

Figure 2: MedGMAE architecture. (a) MedGMAE pre-training framework that processes patchified and masked input through an encoder-decoder architecture to predict 3D Gaussian parameters, which are then rendered and optimized via reconstruction loss. (b) Extended MedGMAE* with multi-level residual blocks for progressive Gaussian parameters refinement.

## 2 RELATED WORK

### 2.1 PATCH-BASED MASKED IMAGE MODELING

Masked autoencoders learn representations by reconstructing masked input regions. MAE (He et al., 2022) uses a ViT encoder (Dosovitskiy et al., 2020) processing only 25% visible patches and a lightweight decoder for reconstruction, enabling efficient pretraining. In medical imaging, existing self-supervised methods focus on different architectural designs and masking strategies to improve representation learning (Zhou et al., 2023b; Xie et al., 2022b; Tang et al., 2024; Tian et al., 2023; Tang et al., 2022a; Goncharov et al., 2023). Despite their architectural variations and different masking mechanisms, all these approaches remain fundamentally constrained by the voxel-level reconstruction objective, which encourages local interpolation rather than global structural understanding

of anatomical features. We propose a fundamentally different approach using 3D Gaussian parameter prediction instead of voxel-level reconstruction. Unlike discrete intensity prediction, our method reconstructs anatomy through continuous geometric primitives, enabling structured representation learning aligned with anatomical continuity.

## 2.2 3D GAUSSIAN SPLATTING FOR MEDICAL IMAGING

3D Gaussian Splatting (3DGS) was developed for rendering 3D natural scenes (Kerbl et al., 2023). This approach has since been applied across diverse medical reconstruction scenarios, including 3D CT reconstruction (Li et al., 2025; Cai et al., 2024; Zha et al., 2024), coronary artery reconstruction (Fu et al., 2024), and 4D CT reconstruction (Fu et al., 2025; Yu et al., 2025).

**Limitations and motivation.** While Gaussian Masked Autoencoders ((Rajasegaran et al., 2025)) pioneered this approach for 2D images—using the z-axis of 3D Gaussians to infer abstract 2.5D layers for spatial understanding—our motivation is fundamentally different. Instead of inferring abstract structure, we leverage the continuous and parameter-efficient nature of 3D Gaussians to holistically represent true 3D anatomical volumes, directly addressing the limitations of discrete voxel models for capturing continuous anatomy. This objective is better suited for downstream 3D tasks like segmentation and registration, and also unlocks a novel application in accelerating CT reconstruction. Our key insight is: leverage 3D Gaussian primitives as intermediate representations for masked autoencoder pre-training, learning geometric structures rather than discrete voxels. This shifts the objective from local reconstruction to geometric reasoning, encouraging spatial reasoning and anatomical priors while addressing initialization challenges in existing 3DGS medical methods.

## 3 METHOD

### 3.1 PRELIMINARIES

**3D Gaussian primitives and volume rendering.** In medical imaging domain, each 3D Gaussian primitives is parameterized by a center position $\mu \in \mathbf{R}^3$ and a covariance matrix $\Sigma \in \mathbf{R}^{3 \times 3}$, which jointly define the spatial distribution and morphology of the Gaussian primitive. In addition, each Gaussian carries an intensity value $I$ that represents the intensity at the Gaussian center. In our implementation, we follow the standard practice of decomposing the covariance matrix $\Sigma = RSS^T R^T$ into a scaling matrix $S = \text{diag}(s) \in \mathbf{R}^{3 \times 3}$ represented by a scale vector $s \in \mathbf{R}^3$, and a rotation matrix $R \in \mathbf{R}^{3 \times 3}$ parameterized by a rotation quaternion $\phi \in \mathbf{R}^4$. Consequently, each Gaussian is represented by an 11-dimensional parameter vector $g = \{\mu, s, \phi, I\} \in \mathbf{R}^{11}$. For volumetric rendering, we reconstruct the complete 3D volume by evaluating the Gaussian field at discrete grid positions corresponding to the target volumetric dimensions. Each Gaussian contribution to any spatial position $X$ is mathematically described by:

$$G_i(X|g_i) = I_i \cdot e^{-\frac{1}{2}(X-\mu_i)^T \Sigma_i^{-1}(X-\mu_i)}, \tag{1}$$

where $X \in \mathbf{R}^3$ denotes a position in the 3D space, and $g_i = \{\mu_i, s_i, \phi_i, I_i\}$ represents the parameters of the $i$-th Gaussian. The exponential term defines the spatial decay of the Gaussian influence based on the Mahalanobis distance from its center, naturally encoding the ellipsoidal shape through the covariance structure. The final volumetric intensity is computed as a spatially-localized aggregation of contributions from nearby Gaussians:

$$V(X|g_i) = \sum_{i:||X-\mu_i|| \leq d_i} G_i(X|g_i), \tag{2}$$

where $d_i$ defines the effective radius of influence for each Gaussian, typically set based on the eigenvalues of the covariance matrix to ensure computational efficiency while maintaining rendering quality. This localized aggregation strategy enables efficient rendering by avoiding computations for Gaussians with negligible contributions, making the differentiable rendering process tractable for large-scale medical volumes.

### 3.2 PROPOSED APPROACH

We propose MedGMAE, a framework that replaces voxel-level reconstruction with 3D Gaussian parameters prediction for medical volumetric representation learning in Fig.2(a).

**MedGMAE representation learning:** The model consists of a Vision Transformer (ViT)-based encoder, a lightweight Transformer decoder, and a differentiable Gaussian renderer specifically designed for volumetric medical data reconstruction. For a given 3D medical image patch with dimensions $96 \times 96 \times 96$, we first patchify it into $N$ non-overlapping patches of size $12 \times 12 \times 12$, resulting in $N = 512$ patches. We then randomly mask these patches with a masking ratio $r$, typically set to 0.75, yielding $n$ visible patches where $n = N \times (1 - r)$. The ViT encoder processes only the visible patches and encodes them from raw patch representations to latent embeddings $x_i \in \mathbf{R}^{d_{enc}}$, where $i \in \{1, 2, 3, \ldots, n\}$. The decoder employs $k$ learnable query tokens $q_j \in \mathbf{R}^{d_{dec}}$, $j \in \{0, 1, 2, \ldots, k-1\}$, where $k$ represents the number of 3D Gaussians to be predicted. Importantly, $k$ can be set to any value independent of the number of masked tokens, providing flexibility in controlling the reconstruction granularity. We project the encoder latent embeddings to the decoder dimension space as $\hat{x}_i \in \mathbf{R}^{d_{dec}}$ and construct the decoder input by concatenating three components: the encoder class token, the learnable Gaussian query tokens, and the remaining encoder tokens:

$$X_{dec} = \{\hat{x}_1\} \cup \{q_j\}_{j=1}^k \cup \{\hat{x}_i\}_{i=2}^n \tag{3}$$

The decoder processes the $X_{dec}$ tokens through multiple Transformer blocks with multi-head self-attention mechanisms. This allows the query tokens to attend to the visible patch embeddings and aggregate spatial-semantic information necessary for accurate 3D Gaussian parameter prediction. The decoder outputs $k$ sets of Gaussian parameters, with each query token predicting one 3D Gaussian primitive through dedicated parameter heads. Each predicted 3D Gaussian is an 11-dimensional vector comprising position coordinates $\mu \in \mathbf{R}^3$, anisotropic scaling factors $s \in \mathbf{R}^3$, rotation quaternion $\phi \in \mathbf{R}^4$, and intensity $I \in \mathbf{R}^1$. The conversion from decoder features to Gaussian parameters is accomplished through four specialized linear prediction heads: a Gaussian center head, a scale head, a rotation head, and an intensity head. Each head applies appropriate activation functions to ensure parameter validity: sigmoid activation for positions and densities to constrain values within [0,1], and L2 normalization for rotation quaternions to maintain unit length. To ensure stable training and balanced parameter distributions across the three spatial dimensions, we employ custom initialization strategies for the prediction heads. All heads utilize Xavier uniform initialization for weights, while biases are initialized specifically for each parameter type: position and rotation heads use zero initialization, the scale head employs a constant bias of -1.386 (resulting in approximately 0.2 after sigmoid activation), and the density head uses a bias of -0.405 (yielding approximately 0.5 after sigmoid activation). This initialization scheme promotes consistent scale distributions across x, y, and z dimensions while providing reasonable starting values for Gaussian intensity.

**Differentiable volumetric rendering and training:** Once we obtain $k$ predicted 3D Gaussians, we employ a differentiable volumetric renderer to reconstruct the 3D medical image. The renderer accumulates the contributions of all Gaussians within the volume space, with each Gaussian influence determined by its spatial extent and intensity. During training, we apply the reconstruction loss only to the originally masked regions, computed as the mean squared error between the rendered volume and the ground truth image. This masked reconstruction objective encourages the model to learn meaningful 3D representations while maintaining computational efficiency.

**Extended MedGMAE for reconstruction:** For enhanced reconstruction performance, we further present MedGMAE* with multi-level residual blocks (a hierarchically extended MedGMAE structure (Hyun & Heo, 2024)), utilizing more Gaussians to capture fine-grained volumetric details in Fig.2(b). We define a hierarchical structure with levels $l \in \{0, 1, 2\}$, from coarse to fine granularity, where each level contains a set of Gaussian parameters. Specifically, we establish dependencies between Gaussian parameters of adjacent levels, where Level 0 contains $N_0$ base Gaussians, Level 1 expands to $m_1 \times N_0$ Gaussians, and Level 2 expands to $m_2 \times N_0$ Gaussians. We model the 3D representation in a coarse-to-fine manner by assigning coarser- and finer-level Gaussians for coarser and finer details. For scale parameters, we enforce hierarchical reduction as:

$$s^l = s^0 + \hat{s}^l \cdot \sigma_{scale} - \Delta s^l \tag{4}$$

where $\sigma_{scale} = 0.1$ controls the residual magnitudes, and $\Delta s^l > 0$ to ensure monotonic scale reduction. $\Delta s^1 = 0.02$ for Level 1 and $\Delta s^2 = 0.05$ for Level 2 are adopted. For other parameters, we compute new positions as: $\mu^l = \mu^0 + \hat{\mu}^l \cdot \sigma_\mu$, where $\hat{\mu}^l$ are the predicted residual position parameters. We define residual transformations as: $I^l = I^0 + \hat{I}^l \cdot \sigma_I$, $\phi^l = \text{normalize}(\phi^0 + \hat{\phi}^l \cdot \sigma_{rot})$ where $\sigma_\mu, \sigma_I, \sigma_{rot}$ control the residual magnitudes. Note that all residual prediction modules adopt tanh activation functions to ensure bounded residual outputs and stable training dynamics. This hierarchical densification enables coarse-to-fine reconstruction while maintaining spatial coherence through

Table 1: Comparison of different methods with different proportions on AMOS (Ji et al., 2022), FLARE'22 (Ma et al., 2024), BTCV (Landman et al., 2015) and SegTHOR (Lambert et al., 2020). The DSC (%) is reported. **val** (bold) / val (underline) : top method / second method. † denotes we utilize official pre-training weights. ‡ denotes the results are copied from VoCo (Wu et al., 2024b).

| Pretrain Method | AMOS | | | FLARE'22 | | | BTCV | | | SegTHOR | | |
|---|---|---|---|---|---|---|---|---|---|---|---|---|
| | 1% | 10% | 100% | 1% | 10% | 100% | 1% | 10% | 100%‡ | 1% | 10% | 100% |
| *Training from scratch* | | | | | | | | | | | | |
| UNETR | 23.67 | 60.06 | 77.02 | 22.47 | 56.46 | 70.81 | 28.05 | 42.85 | 79.82 | 42.31 | 72.72 | 85.82 |
| SwinUNETR | 28.94 | 63.45 | 82.51 | 35.89 | 63.38 | 75.38 | 27.71 | 51.33 | 80.53 | 44.82 | 73.93 | 87.35 |
| *General self-supervised methods* | | | | | | | | | | | | |
| SparK | 36.14 | 71.68 | 84.07 | 36.48 | 71.74 | 80.67 | 30.69 | 51.26 | - | 44.76 | 80.36 | 88.08 |
| MAE | 54.67 | 72.94 | 83.61 | 62.35 | 77.01 | 82.56 | 62.04 | 75.01 | - | 66.72 | 83.60 | 88.52 |
| *Medical self-supervised methods* | | | | | | | | | | | | |
| MG† | 25.72 | 46.94 | 62.99 | 27.30 | 48.18 | 57.33 | 29.27 | 38.04 | 81.45 | 36.96 | 60.16 | 83.79 |
| TransVW† | 18.72 | 66.91 | 82.58 | 4.81 | 62.07 | 75.78 | 5.63 | 8.42 | - | 8.91 | 31.30 | 87.46 |
| UniMiSS† | 29.49 | 66.34 | 79.92 | 24.92 | 60.99 | 74.71 | 32.95 | 47.08 | - | 42.92 | 76.59 | 84.34 |
| SUP† | 25.60 | 64.95 | 82.45 | 33.72 | 60.35 | 74.96 | 28.75 | 49.67 | 81.54 | 41.74 | 73.46 | 87.22 |
| PCRLv2† | 21.07 | 39.07 | 54.14 | 27.71 | 42.97 | 54.29 | 24.01 | 30.48 | 81.74 | 40.22 | 74.71 | 85.77 |
| GVSL† | 24.25 | 63.45 | 81.38 | 26.33 | 59.54 | 73.27 | 24.86 | 41.79 | 81.87 | 42.56 | 77.40 | 86.98 |
| vox2vec† | 32.76 | 62.30 | 74.78 | 34.11 | 61.99 | 70.33 | 35.29 | 51.77 | - | 47.21 | 73.98 | 86.77 |
| HySparK† | 34.50 | 64.32 | **85.58** | 37.54 | 73.60 | 82.35 | 35.81 | 51.54 | - | 58.81 | **83.95** | 88.74 |
| VoCo† | 55.81 | 73.34 | 84.44 | 57.66 | **78.84** | 83.12 | **73.20** | **77.85** | **83.85** | 67.12 | 83.87 | 88.70 |
| MedGMAE | **58.79** | **75.65** | 84.90 | **62.72** | 78.72 | **83.77** | 66.19 | 77.11 | 83.22 | **70.92** | 83.91 | **89.15** |

base Gaussian constraints, significantly enhancing the model's ability to capture fine-grained details in CT reconstruction.

# 4 EXPERIMENTS

## 4.1 DATASETS

**Pre-training datasets.** For self-supervised pre-training, we utilize the AbdomenAtlas1.0Mini dataset (Li et al., 2024a), which contains 5,195 CT scans. All scans are first resampled spacing of 1.5mm×1.5mm× 1.5mm using trilinear interpolation. The Hounsfield Unit (HU) values are then clipped to the range [-175, 250]. Finally, the intensity values are normalized to the range [0, 1].

**Downstream datasets.** For segmentation tasks, we conduct experiments on four public datasets: AMOS (Ji et al., 2022), FLARE'22 (Ma et al., 2024), BTCV (Landman et al., 2015), and SegTHOR (Lambert et al., 2020), with official training-validation split with 1%, 10% and 100% proportions. Medical image classification tasks are conducted on the CT-RATE dataset (Hamamci et al., 2024) with official data partition. For registration tasks we perform experiments on IXI (Kim et al., 2021) and OASIS (Marcus et al., 2007) with same data split as (Wu et al., 2024a). Also, CT reconstruction experiments are conducted on the low-dose Chest and Abdomen CT: AAPM-Mayo dataset (Moen et al., 2021).

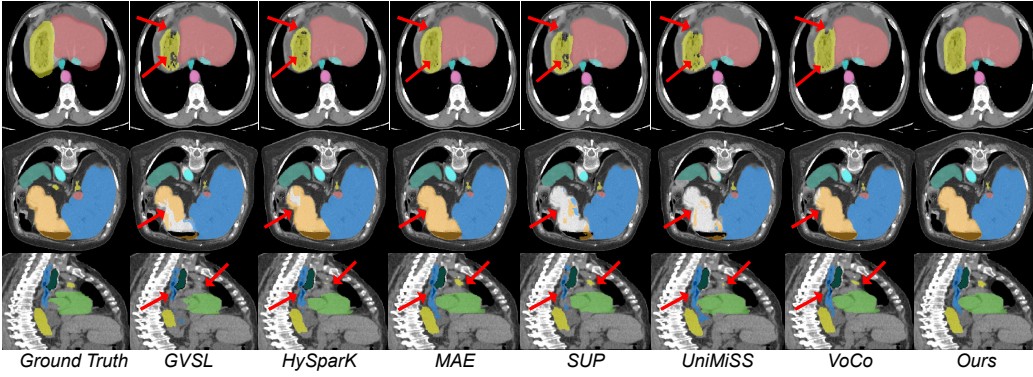

Figure 3: Visualization of one-shot segmentation results for AMOS (row 1), FLARE'22 (row 2) and SegTHOR (row 3).

## 4.2 IMPLEMENTATION DETAILS

For pre-training, we sample the pre-training volumes into $96 \times 96 \times 96$ patches using RandCrop-ByPosNegLabeld with 3 crops per volume. We use the AdamW optimizer, an initial learning rate of 1e-5, and a cosine-annealing scheduler for all experiments. The pre-training uses a batch size of 8 and trains the model for about 100K steps. All experiments use a fixed random seed of 41 to ensure reproducibility. We evaluate our method using task-specific metrics: Dice Similarity Coefficient (DSC) for segmentation, Area Under the Curve (AUC) for classification, DSC for registration, and Peak Signal-to-Noise Ratio (PSNR) and Structural Similarity Index Measure (SSIM) for reconstruction tasks.

For downstream transfer tasks, we adopt UNETR (Hatamizadeh et al., 2022) as the baseline network followed as (Chen et al., 2023; Tang et al., 2024). For segmentation task, all the pre-processing strategies are the same as (Tang et al., 2022b). For classification task, We resample the volume to $1.5 \times 1.5 \times 3.0$ mm, clip the HU range to $[-1000, 1000]$ and rescale it to [0, 1]. The volume size is set to be $192 \times 192 \times 96$. The model is trained for 100 epochs with a batch size of 96, using AdamW as the optimizer with a learning rate of 3e-2 and weight decay of 0.05. For registration task, our registration algorithm based on TransMorph (Chen et al., 2022) and all the registration pre-processing and training strategies are the same as (Wu et al., 2024a). All experiments are conducted on NVIDIA H20 GPUs.

For CT reconstruction task, we follow the experimental setup of (Li et al., 2025). The key difference is that FBP reconstruction results are cropped using non-overlapping sliding windows and fed into MedGMAE as input. The output Gaussian parameters undergo volume rendering and are concatenated back to original size. To reduce FBP artifact interference, we conduct experiments with 80, 120, and 160 projections. All experiments are trained for 15,000 iterations on Nvidia 3090 GPUs. We evaluate the original 3DGR, 3DGR with MedGMAE initialization, and 3DGR with MedGMAE* initialization using training time, iterations required to reach PSNR=35 and SSIM=90%, and final PSNR and SSIM after complete training.

**Comparison with state-of-the-art methods.** We select both general and medical self-supervised learning methods for comprehensive comparison. Following (Wu et al., 2024b), we select UNETR (Hatamizadeh et al., 2022), SwinUNETR (Tang et al., 2022a) as compared baseline model. For segmentation tasks, we compare against prominent masked image modeling (MIM) methods, including MAE (He et al., 2022; Chen et al., 2023) and SparK (Tian et al., 2023), under identical experimental settings. Additionally, we select nine recent and well-known self-supervised methods: Models Genesis (MG) (Zhou et al., 2021), TransVW (Haghighi et al., 2021), UniMiSS (Xie et al., 2022a), Swin UNETR Pretrained method (SUP) (Tang et al., 2022a), PCRLv2 (Zhou et al., 2023a), GVSL (He et al., 2023), vox2vec (Goncharov et al., 2023), HySparK (Tang et al., 2024), and VoCo (Wu et al., 2024b). For registration tasks, we compare against methods trained from scratch, including VoxelMorph (Balakrishnan et al., 2019), TransMorph (Chen et al., 2022), and SwinUNETR (Hatamizadeh et al., 2021), as well as methods with pre-training such as SUP (Tang et al., 2022b), SuPreM (Li et al., 2024b), and VoCo (Wu et al., 2024b). To ensure fair comparison, we utilize official implementations and loaded official pre-trained weights for all medical SSL methods before fine-tuning.

## 5 RESULTS

### 5.1 PROMISING DOWNSTREAM TRANSFER RESULTS

**Medical image segmentation**. Following previous work, we fine-tuned pre-trained models using 1%, 10%, and 100% of the training data on AMOS, FLARE'22, BTCV, and SegTHOR datasets, respectively. The segmentation results are presented in Table 1. MedGMAE achieves the best or second-best DSC scores among all compared methods across different data regimes. Notably, our method demonstrates particularly strong performance in low-data scenarios, outperforming the previous best method VoCo by 2.98% and 5.06% on AMOS and FLARE'22, respectively, with 1% data. Compared to training from scratch baselines (Hatamizadeh et al., 2022; 2021), our pre-trained MedGMAE demonstrates substantial improvements, with gains of 20-35% in 1% data scenarios across all datasets. Even with full training data, pre-training consistently provides meaningful improvements of 2-8% over the corresponding from-scratch methods. These results demonstrate that

Table 2: Performance comparison on CT-RATE dataset for classification task. The AUC (%) is shown. **val** (bold) / val (underline) : top method / second method. † denotes official pre-training weights.

| | Method | AUC |
|---|---|---|
| *Scratch* | UNETR | 71.43 |
| | SwinUNETR | 74.29 |
| *Fine-tuning* | VoCo-10K† | 72.11 |
| | VoCo-160K† | 76.02 |
| | SUP† | 76.04 |
| | MedGMAE | **76.40** |

Table 3: The DSC(%) of registration on IXI and OASIS datasets. ‡ denotes the results are copied from VoCo (Wu et al., 2024a). The best results are in **bold**.

| Method | IXI | OASIS |
|---|---|---|
| *Training From Scratch* | | |
| VoxelMorph† | 71.5 | 78.6 |
| TransMorph† | **74.5** | 81.6 |
| SwinUNETR† | 72.6 | 81.8 |
| *Fine-tuning* | | |
| SUP† | 67.7 | 81.5 |
| SuPreM† | 72.9 | 81.2 |
| VoCo† | 73.6 | 84.4 |
| MedGMAE | 73.7 | **85.7** |

Table 4: Comprehensive reconstruction comparison across different projection views. 3DGR refers to the initialization method employed in the original paper (Li et al., 2025), whereas MedGMAE and MedGMAE* indicate initialization using Gaussian points estimated through zero-shot inference by our proposed model. Values are reported as mean ± standard deviation. The best results are in **bold**.

| Method | Time(min) | iter(P=35) | iter(S=90%) | PSNR(full) | SSIM(full) |
|---|---|---|---|---|---|
| | | | 80 projections | | |
| 3DGR | $397_{\pm 39.5}$ | $1670_{\pm 371.6}$ | $1140_{\pm 145.7}$ | $\mathbf{44.6}_{\pm 1.19}$ | $\mathbf{98.4}_{\pm 0.32}$ |
| MedGMAE | $303_{\pm 30.9}$ | $1135_{\pm 95.0}$ | $1100_{\pm 102.5}$ | $43.9_{\pm 1.01}$ | $97.5_{\pm 0.51}$ |
| MedGMAE* | $\mathbf{251}_{\pm 19.8}$ | $\mathbf{990}_{\pm 137.5}$ | $\mathbf{820}_{\pm 60.0}$ | $44.1_{\pm 0.97}$ | $98.0_{\pm 0.38}$ |
| | | | 120 projections | | |
| 3DGR | $507_{\pm 47.8}$ | $1660_{\pm 382.6}$ | $1150_{\pm 206.5}$ | $45.2_{\pm 1.49}$ | $98.7_{\pm 0.32}$ |
| MedGMAE | $357_{\pm 22.0}$ | $1040_{\pm 91.7}$ | $980_{\pm 60.0}$ | $\mathbf{46.2}_{\pm 1.17}$ | $98.5_{\pm 0.29}$ |
| MedGMAE* | $\mathbf{335}_{\pm 20.4}$ | $\mathbf{920}_{\pm 74.8}$ | $\mathbf{780}_{\pm 32.0}$ | $45.8_{\pm 1.15}$ | $\mathbf{98.7}_{\pm 0.27}$ |
| | | | 160 projections | | |
| 3DGR | $594_{\pm 140.5}$ | $1711_{\pm 449.8}$ | $1137_{\pm 211.8}$ | $45.1_{\pm 1.53}$ | $98.7_{\pm 0.34}$ |
| MedGMAE | $373_{\pm 20.5}$ | $1055_{\pm 85.7}$ | $967_{\pm 69.9}$ | $\mathbf{46.8}_{\pm 1.36}$ | $98.7_{\pm 0.25}$ |
| MedGMAE* | $388_{\pm 36.6}$ | $\mathbf{960}_{\pm 96.8}$ | $\mathbf{780}_{\pm 33.1}$ | $45.8_{\pm 1.39}$ | $98.7_{\pm 0.21}$ |

our method could learn a strong anatomical representation by using Gaussian representation. Fig. 3 shows the visualization results.

**Medical image classification**. Table 2 presents the performance comparison on the CT-RATE dataset. Compared to training from scratch, MedGMAE shows substantial improvements over the best scratch-trained baseline Swin-Bv2 by 2.11%. Among pre-trained methods, MedGMAE surpasses the previous best performers VoCo-160K and SUP by 0.38% and 0.36% respectively, demonstrating the effectiveness of our pre-training approach.

**Medical image registration**. Table 3 presents the DSC performance comparison on IXI and OASIS datasets for medical image registration tasks. MedGMAE achieves the best performance on OASIS and competitive results on IXI. Compared to the best scratch-trained baselines, our method provides substantial improvements of 1.1% on IXI and 3.9% on OASIS. Among pre-trained methods, MedGMAE outperforms the previous state-of-the-art VoCo by 1.3% on OASIS, confirming the effectiveness of our pre-training approach for medical image registration tasks. It worth noting that both IXI and OASIS are from *unseen* MRI modality, which demonstrates the generalization ability of MedGMAE.

## 5.2 GEOMETRY-AWARE ZERO-SHOT INITIALIZATION FOR 3DGS-BASED MEDICAL IMAGE RECONSTRUCTION

As shown in Table 4 and Fig. 4, MedGMAE demonstrates significant acceleration in training convergence across all projection settings. For training efficiency, MedGMAE reduces training time

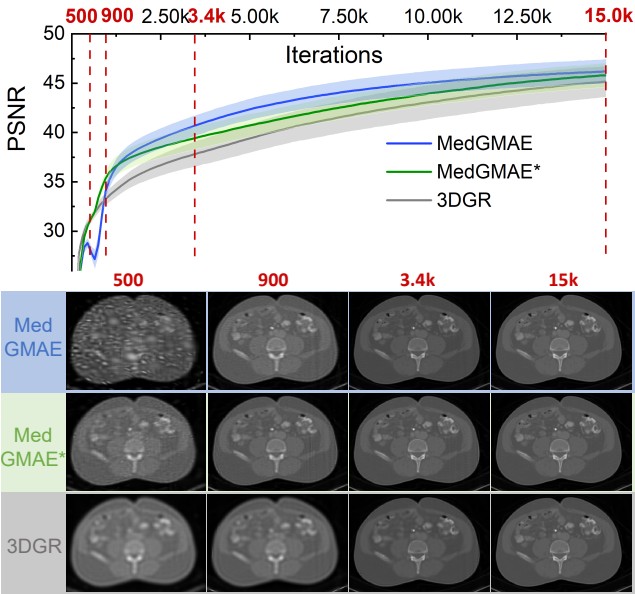

Figure 4: CT reconstruction convergence analysis on AAPM-Mayo dataset. Top: Average PSNR curves with standard error bands showing reconstruction quality improvement over training iterations for different methods. Bottom: Visual comparison of reconstructed CT slices at 500, 900, 3.4k, and 15k iterations for MedGMAE, MedGMAE*, and 3DGR methods, demonstrating the faster convergence and superior reconstruction quality of our approaches.

by 31.0%, 35.0%, and 37.2% compared to 3DGR baseline with 80, 120, and 160 projections respectively. More importantly, MedGMAE substantially accelerates convergence speed, requiring 39.4% and 28.1% fewer iterations to reach PSNR=35 and SSIM=90% benchmarks on average. The residual-extended MedGMAE* further improves convergence performance, achieving even faster iteration counts for quality thresholds while maintaining comparable final reconstruction quality. These results demonstrate that our pre-training approach significantly enhances training efficiency for 3D Gaussian representation-based CT reconstruction without compromising final image quality. Statistical analysis using t-tests revealed that our proposed MedGMAE initialization methods significantly outperformed 3DGR ($p < 0.001$) in traiing efficiency.

### 5.3 ABLATION STUDY

Table 5 presents the ablation study results on MedGMAE components across three segmentation datasets. Adding voxel-based SSL provides substantial improvements of 6-12% over the baseline. Our proposed Gaussian-based SSL further enhances performance by 1-2% compared to voxel-based approaches, confirming the superiority of 3D Gaussian representation over voxel-based reconstruction.

Table 5: Transfer ablation on MedGMAE. The DSC (%) is reported.

| Proxy | | SSL | AMOS | FLARE'22 | SegTHOR |
|---|---|---|---|---|---|
| Voxel | Gaussian | | | | |
| | | | 77.02 | 70.81 | 85.82 |
| ✓ | | ✓ | 83.61 | 82.56 | 88.52 |
| | ✓ | ✓ | 84.90 | 83.77 | 89.15 |

## 6 CONCLUSION

In this paper, we present MedGMAE, a novel self-supervised pre-training framework that replaces voxel-level reconstruction with 3D Gaussian representation. Leveraging the more efficient and continuous 3D Gaussian primitives, MedGMAE achieves promising encoder transfer performance on diverse downstream tasks including segmentation, classification, and registration. Besies, the transferable decoder enables a $1.39\times$ acceleration compared to original 3DGR-CT reconstruction methods. Extensive experimental results demonstrate the effectiveness of MedGMAE across multiple medical imaging applications. However, in CT reconstruction tasks, the result are affected by noise from FBP reconstruction, which could be improved by training a multi-view 3D Gaussian foundation model.

## ACKNOWLEDGMENTS

Supported by National Natural Science Foundation of China under Grant 62271465, Suzhou Basic Research Program under Grant SYG202338, and Jiangsu Province Science Foundation for Youths (NO. BK20240464).

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

## 7 APPENDIX

### 7.1 THE USE OF LARGE LANGUAGE MODELS

Large language models (LLMs) were employed solely to improve the clarity and readability of the manuscript. They were not involved in the conception of the study, the design or execution of experiments, the analysis or interpretation of results, or any other scientific aspect of this work.

### 7.2 ETHICS STATEMENT

This work involves the analysis of human CT scan data exclusively sourced from publicly available datasets. We acknowledge that the representativeness and potential biases present in these public datasets may influence the fairness and generalizability of our proposed model. We encourage future work to validate our methods on more diverse and representative datasets to ensure equitable healthcare outcomes. All datasets used in this study were previously collected with appropriate ethical approvals and consent procedures as documented by the original data contributors. Beyond these considerations, we have identified no additional ethical conflicts or concerns related to this research.

### 7.3 REPRODUCIBILITY STATEMENT

To ensure the reproducibility of our work, we have made the following efforts: All architectural details, hyperparameters, and training procedures for our proposed method are comprehensively described in Section 3.2 and Section 4.2 of the main paper. For comparative baseline results, we have directly reported performance metrics from the original publications to avoid potential inconsistencies that may arise from reimplementation differences, with proper citations provided throughout. Upon acceptance of this paper, we commit to releasing the complete source code, including training scripts, model implementations, and evaluation, to facilitate full reproducibility of our results.

### 7.4 NETWORK ARCHITECTURE CONFIGURATION

**Encoder Architecture (ViT Large)**

Our MedGMAE employs a Vision Transformer (ViT) Large configuration as the encoder backbone. The detailed specifications are presented in Table 6.

The encoder processes 3D medical images through the following pipeline:

**Decoder Architecture (Lightweight Design)**

The decoder employs a lightweight Transformer architecture optimized for Gaussian parameter prediction, as detailed in Table 7.

**Decoder Input Token Composition**

Table 6: ViT Large Encoder Configuration Details

| Component | Configuration |
|---|---|
| Embedding Dimension | 1536 |
| Number of Attention Heads | 16 |
| Number of Transformer Layers | 12 |
| MLP Ratio | 4.0 |
| Patch Size | $16 \times 16 \times 16$ or $12 \times 12 \times 12$ |
| Input Image Size | $96 \times 96 \times 96$ |
| Number of Patches | 512 (for $12^3$) or 216 (for $16^3$) |
| Dropout Rate | 0.0 |
| Attention Dropout Rate | 0.0 |
| Drop Path Rate | 0.1 |

Table 7: Gaussian Decoder Configuration Details

| Component | Configuration |
|---|---|
| Embedding Dimension | 528 |
| Number of Attention Heads | 16 |
| Number of Transformer Layers | 8 |
| MLP Ratio | 4.0 |
| Number of Gaussian Query Tokens | 512 |
| Encoder-to-Decoder Projection | $1536 \rightarrow 528$ Linear Layer |
| Dropout Rate | 0.0 |
| Attention Dropout Rate | 0.0 |
| Drop Path Rate | 0.1 |

The decoder processes a carefully constructed sequence of tokens:

$$\mathbf{X}_{dec} = \{\mathbf{x}_{cls}\} \cup \{\mathbf{q}_j\}_{j=1}^{512} \cup \{\mathbf{x}_i\}_{i=2}^{n} \tag{5}$$

where:

- $\mathbf{x}_{cls}$: Class token from encoder (1 token)
- $\{\mathbf{q}_j\}_{j=1}^{512}$: Gaussian query tokens (512 tokens)
- $\{\mathbf{x}_i\}_{i=2}^{n}$: Remaining visible patch tokens (127 tokens for 75% masking)

Total decoder input length: $1 + 512 + 127 = 640$ tokens.

### 7.5 GAUSSIAN PARAMETER PREDICTION HEADS

**Four Specialized Prediction Heads**

Each Gaussian is parameterized by an 11-dimensional vector comprising position, scale, rotation, and intensity. Four specialized linear heads predict these parameters:

Table 8: Gaussian Parameter Prediction Head Specifications

| Parameter | Dimension | Activation | Range | Bias Init |
|---|---|---|---|---|
| Position ($\boldsymbol{\mu}$) | 3 | Sigmoid | $[0,1]^3$ | 0.0 |
| Scale ($\mathbf{s}$) | 3 | Sigmoid | $[0,1]^3$ | -1.386 |
| Rotation ($\phi$) | 4 | L2 Normalize | Unit Quaternion | 0.0 |
| Density ($\alpha$) | 1 | Sigmoid | $[0,1]$ | -0.405 |

**Custom Initialization Strategy**

To ensure balanced parameter distributions across spatial dimensions, we employ specialized initialization:

$$\text{Position Head:} \quad \mathbf{W} \sim \mathcal{U}(-\sqrt{6/d}, \sqrt{6/d}), \quad \mathbf{b} = \mathbf{0} \tag{6}$$

$$\text{Scale Head:} \quad \mathbf{W} \sim \mathcal{U}(-\sqrt{6/d}, \sqrt{6/d}), \quad \mathbf{b} = -1.386 \tag{7}$$

$$\text{Rotation Head:} \quad \mathbf{W} \sim \mathcal{U}(-\sqrt{6/d}, \sqrt{6/d}), \quad \mathbf{b} = \mathbf{0} \tag{8}$$

$$\text{Density Head:} \quad \mathbf{W} \sim \mathcal{U}(-\sqrt{6/d}, \sqrt{6/d}), \quad \mathbf{b} = -0.405 \tag{9}$$

The bias initialization ensures reasonable starting distributions:

- Scale bias of -1.386 results in $\sigma(-1.386) \approx 0.2$ after sigmoid activation

- Density bias of -0.405 results in $\sigma(-0.405) \approx 0.5$ after sigmoid activation

## 7.6 DIFFERENTIABLE GAUSSIAN RENDERING ALGORITHM

### CUDA Implementation Details

Our CUDA implementation employs several optimization strategies:

CUDA Gaussian Rendering Kernel [1] Gaussian parameters $\{\boldsymbol{\mu}_i, \mathbf{s}_i, \boldsymbol{\phi}_i, \alpha_i\}_{i=1}^N$ Grid points $\{\mathbf{x}_j\}_{j=1}^M$, Pixel mask $\mathbf{M}$ Rendered intensity grid $\mathbf{I}$ Initialize shared memory buffers for covariance matrices and centers gaussian_idx = atomicAdd(work_counter, 1) $< N$ Load Gaussian parameters into shared memory Compute bounding box using $2\sigma$ rule: $\text{expand}_d = 2.0 \times s_{i,d} \times \text{grid\_size}_d$ for $d \in \{x, y, z\}$ $\text{bounds}_d = [\mu_{i,d} - \text{expand}_d, \mu_{i,d} + \text{expand}_d]$ each voxel $\mathbf{x}_j$ in bounding box $\mathbf{M}[j] = 1$ (masked region) Compute $\Delta\mathbf{x} = \mathbf{x}_j - \boldsymbol{\mu}_i$ Compute power $= -0.5\Delta\mathbf{x}^T\boldsymbol{\Sigma}_i^{-1}\Delta\mathbf{x}$ intensity $= \alpha_i \exp(\text{power})$ atomicAdd($\mathbf{I}[j]$, intensity)

### Sparse Rendering Optimization

For masked regions, we implement sparse rendering that only computes intensities for required pixels:

[h!] Sparse Gaussian Rendering [1] Sparse grid points $\{\mathbf{x}_j\}_{j=1}^M$ (only masked pixels) Gaussian parameters $\{\boldsymbol{\mu}_i, \mathbf{s}_i, \boldsymbol{\phi}_i, \alpha_i\}_{i=1}^N$ Sparse intensity values $\{I_j\}_{j=1}^M$ each sparse point $j$ in parallel $I_j = 0$ each Gaussian $i$ Check if point $\mathbf{x}_j$ within $2\sigma$ bounds of Gaussian $i$ within bounds Compute intensity contribution and add to $I_j$

This sparse approach reduces computational complexity from $O(N \times H \times W \times D)$ to $O(N \times M)$ where $M$ is the number of masked pixels (typically $0.75 \times H \times W \times D$).

## 7.7 TRAINING CONFIGURATION

The training parameters are shown in Table 9.

## 7.8 ADDITIONAL EXPERIMENTAL RESULTS

### Downstream Classification Performance on CT-RATE Dataset

### CT Reconstruction Performance Analysis

Figure 6 demonstrates the superior performance of MedGMAE in accelerating CT reconstruction convergence. Our method shows significant improvements across different projection views (80, 120, and 160 projections), with MedGMAE achieving faster convergence and better reconstruction quality compared to the baseline 3DGR method.

Table 9: Training Hyperparameters

| Parameter | Value |
|---|---|
| Batch Size | 8 |
| Learning Rate | $1 \times 10^{-5}$ |
| Weight Decay | 0.05 |
| Optimizer | AdamW |
| Learning Rate Schedule | Cosine Annealing |
| Warmup Steps | 2000 |
| Max Training Steps | 100000 |
| Gradient Clipping | 1.0 |
| **Gaussian Parameters** | |
| Number of Gaussians | 512 |
| Maximum Scale | 0.5 |
| Temperature ($\tau$) | 0.5 |

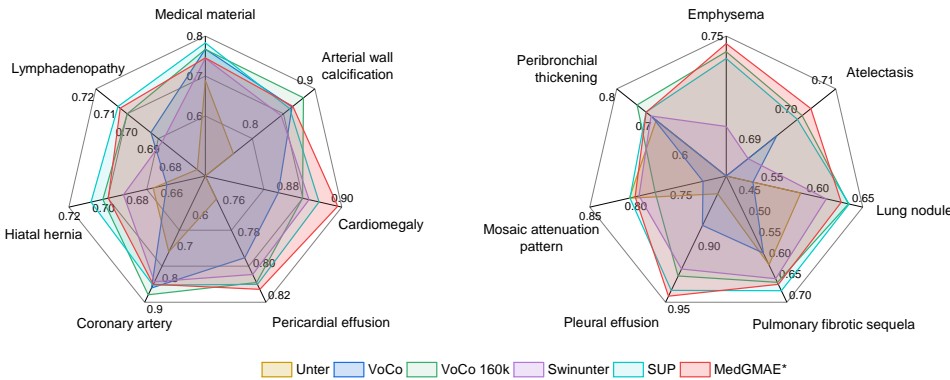

Figure 5: Classification performance comparison on CT-RATE dataset. The radar charts show the Area Under Curve (AUC) scores for different disease categories.

Table 10: Ablation study on pre-training mask ratio for Segmentation tasks. All models use K=512 Gaussian points. rL2 error is computed on reconstruction quality. AMOS and SegTHOR show segmentation Dice scores (%). The best results are in **bold**.

| Mask Ratio | rL2 error | AMOS(100%) | SegTHOR(100%) |
|---|---|---|---|
| 25% | **0.0031** | 80.03 | 86.03 |
| 50% | 0.0068 | 79.01 | 86.00 |
| 75% | 0.0117 | 84.90 | 89.15 |
| 85% | 0.0147 | **85.17** | **89.17** |

Table 11: Ablation study on pre-training Gaussian Numbers (K) for Segmentation tasks. All models use pre-training mask ratio of 75%. rL2 error is computed on reconstruction quality. AMOS and SegTHOR show segmentation Dice scores (%). The best results are in **bold**.

| K | rL2 error | AMOS(100%) | SegTHOR(100%) |
|---|---|---|---|
| 256 | 0.0143 | 79.99 | 85.82 |
| 512 | **0.0117** | **84.90** | **89.15** |
| 768 | 0.0144 | 79.33 | 85.40 |
| 1024 | 0.0137 | 78.87 | 85.89 |

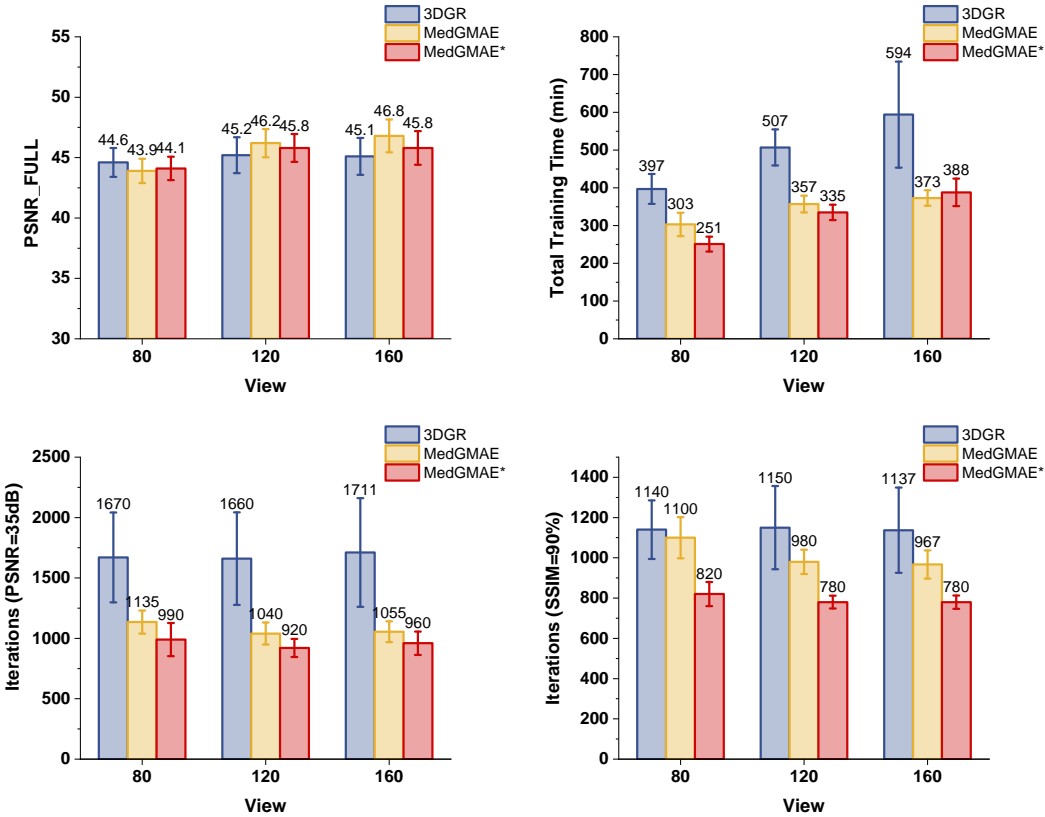

Figure 6: CT reconstruction performance comparison across different projection views. Top row shows PSNR convergence, training time, iterations to reach PSNR=35dB, and iterations to reach SSIM=90%. Bottom row presents the convergence curves and visual reconstruction quality at different iteration stages (500, 900, 3.4k, 15k iterations) for MedGMAE, MedGMAE*, and 3DGR methods.

Table 12: Ablation study on pre-training mask ratio for zero-shot inference reconstruction. All models use K=512 Gaussian points. Values are reported as mean $\pm$ standard deviation. The best results are in **bold**.

| Mask Ratio | Time(min) | iter(P≥35) | iter(S≥90%) | PSNR(full) | SSIM(full) |
|---|---|---|---|---|---|
| 25% | $541.5_{\pm 60.3}$ | $1250.0_{\pm 158.1}$ | $1050.0_{\pm 70.7}$ | $44.6_{\pm 2.02}$ | $98.4_{\pm 0.39}$ |
| 50% | $572.4_{\pm 38.2}$ | $1220.0_{\pm 122.9}$ | $1040.0_{\pm 51.6}$ | $45.3_{\pm 1.31}$ | $98.5_{\pm 0.30}$ |
| 75% | $\mathbf{357.0}_{\pm 22.0}$ | $\mathbf{1040.0}_{\pm 91.7}$ | $\mathbf{980.0}_{\pm 60.0}$ | $\mathbf{46.2}_{\pm 1.17}$ | $\mathbf{98.5}_{\pm 0.29}$ |
| 85% | $545.9_{\pm 56.3}$ | $1050.0_{\pm 85.0}$ | $1000.0_{\pm 47.1}$ | $46.0_{\pm 1.36}$ | $98.3_{\pm 0.38}$ |

Table 13: Ablation study on pre-training Gaussian Numbers (K) for zero-shot inference reconstruction. All models use pre-training mask ratio of 75%. Values are reported as mean $\pm$ standard deviation. The best results are in **bold**.

| K | Time(min) | iter(P≥35) | iter(S≥90%) | PSNR(full) | SSIM(full) |
|---|---|---|---|---|---|
| 256 | $498.8_{\pm 52.9}$ | $1220.0_{\pm 113.5}$ | $1080.0_{\pm 91.9}$ | $45.8_{\pm 1.44}$ | $98.4_{\pm 0.48}$ |
| 512 | $\mathbf{357.0}_{\pm 22.0}$ | $\mathbf{1040.0}_{\pm 91.7}$ | $980.0_{\pm 60.0}$ | $\mathbf{46.2}_{\pm 1.17}$ | $\mathbf{98.5}_{\pm 0.29}$ |
| 768 | $488.4_{\pm 49.8}$ | $1140.0_{\pm 107.5}$ | $1040.0_{\pm 69.9}$ | $45.1_{\pm 0.99}$ | $98.3_{\pm 0.32}$ |
| 1024 | $546.7_{\pm 78.7}$ | $1070.0_{\pm 133.7}$ | $\mathbf{920.0}_{\pm 63.2}$ | $45.2_{\pm 1.20}$ | $98.5_{\pm 0.34}$ |

## 7.9 HYPERPARAMETER SENSITIVITY ANALYSIS

To address concerns about parameter selection and validate the robustness of our approach across different configurations, we conduct systematic ablation studies on two critical hyperparameters: mask ratio and Gaussian primitive count.

**Effect of Mask Ratio.** Tables 10 and 12 jointly investigate the impact of mask ratio on both downstream transfer learning and zero-shot CT reconstruction capabilities. We evaluate four mask ratios (25%, 50%, 75%, 85%) while fixing the number of Gaussian primitives at K=512.

For downstream segmentation tasks (Table 10), the reconstruction quality, measured by reconstruction L2 (rL2) error on masked regions, naturally degrades as the mask ratio increases—from 0.0031 at 25% to 0.0147 at 85% masking. However, downstream performance on both AMOS and SegTHOR tells a different story: segmentation scores improve significantly from 25% to 75% masking (AMOS: 80.03%→84.90%, SegTHOR: 86.03%→89.15%), then plateau at 85%. This trend demonstrates that higher mask ratios force the model to learn more robust and generalizable representations rather than merely memorizing local patterns. Notably, while lower mask ratios (25%, 50%) achieve better reconstruction metrics, they fail to learn representations that transfer effectively to downstream tasks.

For zero-shot CT reconstruction (Table 12), 75% masking achieves the best overall balance: it requires significantly shorter training time and fewer iterations to reach quality thresholds (PSNR$\geq$35dB and SSIM$\geq$90%) compared to lower mask ratios, while maintaining competitive final reconstruction quality (PSNR: 46.2$\pm$1.17, SSIM: 98.5$\pm$0.29). Lower mask ratios require longer training despite achieving marginally better final metrics, suggesting they learn less efficient initialization priors. The 85% mask ratio converges slightly faster to SSIM$\geq$90% but results in lower final quality, indicating potential under-training of the decoder during pre-training. These results consistently demonstrate that 75% masking provides the optimal balance across both transfer learning and reconstruction tasks, aligning with findings in natural image MAE methods.

**Effect of Gaussian Primitive Number.** Tables 11 and 13 examine how the number of Gaussian primitives K affects both transfer performance and reconstruction initialization. We test four configurations (K = 256, 512, 768, 1024) while maintaining the 75% mask ratio.

For downstream segmentation (Table 11), reconstruction quality (rL2 error) shows relatively stable performance across different K values (0.0137–0.0144), suggesting that even moderate numbers of Gaussian primitives can adequately capture volumetric anatomy. However, downstream performance exhibits a clear preference for K=512, achieving the best results on both AMOS (84.90%) and SegTHOR (89.15%), outperforming both smaller and larger configurations. This non-monotonic relationship suggests that K=512 provides an optimal balance: sufficient capacity to model anatomical complexity without introducing excessive parameters.

For zero-shot CT reconstruction (Table 13), K=512 consistently demonstrates superior efficiency and quality: it achieves the fastest training convergence and reaches quality thresholds with fewer iterations, while delivering the best final reconstruction metrics (PSNR: 46.2$\pm$1.17, SSIM: 98.5$\pm$0.29). Smaller configurations (K=256) require longer training and more iterations despite achieving similar final quality, suggesting insufficient geometric prior learning. Larger configurations (K=768, 1024) also show degraded efficiency despite having more representational capacity. This counter-intuitive result indicates that excessive Gaussian primitives may introduce redundancy and optimization challenges that outweigh the benefits of increased capacity. The superior performance of K=512 across both tasks confirms that our method has learned an effective balance between geometric expressiveness and computational efficiency.

**Key Findings.** These comprehensive ablation studies validate our design choices and demonstrate the robustness of MedGMAE across different configurations for the datasets that we used during experiments: (1) The 75% mask ratio consistently emerges as near optimal for both transfer learning and zero-shot reconstruction, balancing reconstruction quality with learned representation quality; (2) "K=512 Gaussian primitives" provides the best efficiency-performance trade-off, avoiding both under-parameterization and over-parameterization; (3) Our method's performance remains stable within reasonable hyperparameter ranges.

## 7.10 RADIAL POWER SPECTRUM ANALYSIS

**Frequency-Domain Analysis.** We performed comprehensive spectral analysis on 50 volumes from the BTCV dataset. For each volume: (1) Ground Truth spectrum: 3D FFT followed by radial binning into 80 frequency bins (normalized frequency range 0-0.5); (2) Voxel MAE reconstruction: obtained using our pre-trained voxel-based masked autoencoder via sliding window inference

(ROI: $96^3$, overlap: 0.5); (3) Gaussian MAE reconstruction: obtained using our pre-trained Gaussian decoder with differentiable splatting-based rendering. All volumes were center-cropped or zero-padded to $96^3$ resolution before FFT computation to ensure consistent frequency binning. We visualized average power spectral density across 10 randomly selected samples with ±1 standard deviation error bands (Fig. 7), and computed spectral L2 distance averaged across all 50 samples (Mean: Voxel=0.125±0.015, Gaussian=0.129±0.018, p=0.12, paired t-test).

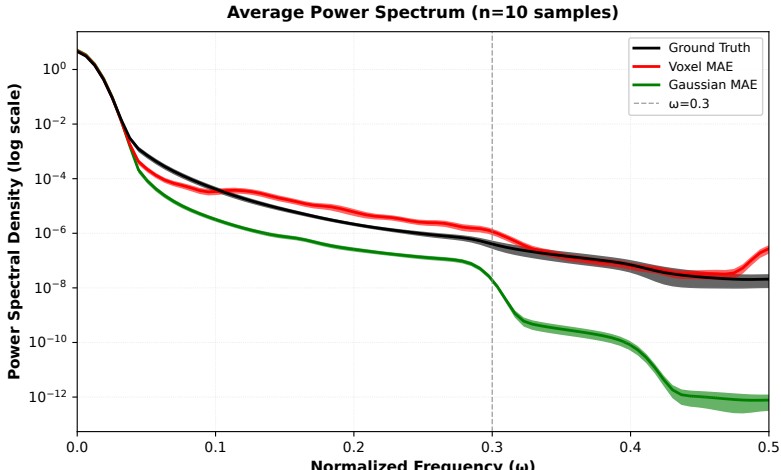

Figure 7: Radial power spectrum comparison across 10 validation samples from BTCV dataset. Average power spectral density curves for Ground Truth (black), Voxel MAE (red), and Gaussian MAE (green) are shown with ± standard deviation error bands (shaded regions). Both reconstruction methods accurately preserve the dominant low-frequency components ($\omega \leq 0.3$), which contain over 99% of the spectral energy. In the high-frequency regime ($\omega > 0.3$, marked by the dashed vertical line), Gaussian MAE demonstrates superior stability with 25% lower variance (std: 0.0031) compared to Voxel MAE (std: 0.0041), exhibiting smoother spectral decay that more closely follows the ground truth distribution. This validates that Gaussian splatting-based parameterization naturally suppresses high-frequency noise artifacts while maintaining anatomical structure fidelity.

### 7.11 Organ-Specific Gaussian Primitive Analysis

To investigate how our Gaussian-based decoder adapts to different anatomical structures, we analyzed the distribution of learned Gaussian primitives across major organ types. We selected eight representative organs from the BTCV dataset covering diverse anatomical characteristics: large solid organs (liver), hollow organs (stomach), paired organs (kidneys), elongated structures (pancreas), tubular structures (aorta), and small organs (gallbladder). For each organ, Gaussian primitives were assigned based on their center positions relative to the ground truth segmentation masks.

Table 14 presents the statistics of Gaussian primitives across 10 validation samples. Several key observations emerge from this analysis:

**Adaptive primitive allocation.** The number of Gaussian primitives scales proportionally with organ volume. The liver, occupying 60.9% of total organ volume, is represented by 1665±747 primitives on average, while the gallbladder (1.0% volume) requires only 26±16 primitives. This demonstrates that our decoder automatically allocates representational capacity according to anatomical complexity without explicit supervision.

**Radius adaptation to organ geometry.** Mean Gaussian radii vary systematically across organ types. Small, compact organs like the gallbladder exhibit larger radii (0.153±0.047), enabling efficient coverage with fewer primitives. In contrast, organs with complex morphology such as the spleen (0.097±0.031) employ smaller, more granular Gaussians to capture fine-grained structural details. This self-adaptive behavior emerges naturally from the reconstruction objective without geometric priors.

Table 14: Gaussian primitive statistics by organ type across 10 BTCV validation samples. Volume percentage is relative to total organ volume per sample. Mean radius and intensity values are averaged across all primitives within each organ region.

| Organ | Volume (% of total) | # Gaussians (count) | Mean Radius (normalized) | Mean Intensity (0–1) |
|---|---|---|---|---|
| Liver | 60.9% | $1665 \pm 747$ | $0.128 \pm 0.006$ | $0.739 \pm 0.052$ |
| Stomach | 16.6% | $502 \pm 332$ | $0.117 \pm 0.014$ | $0.666 \pm 0.109$ |
| Spleen | 6.3% | $134 \pm 103$ | $0.097 \pm 0.031$ | $0.469 \pm 0.079$ |
| Right Kidney | 4.8% | $148 \pm 71$ | $0.139 \pm 0.013$ | $0.770 \pm 0.057$ |
| Left Kidney | 4.6% | $175 \pm 73$ | $0.101 \pm 0.013$ | $0.514 \pm 0.077$ |
| Aorta | 3.5% | $90 \pm 109$ | $0.126 \pm 0.025$ | $0.753 \pm 0.095$ |
| Pancreas | 2.4% | $82 \pm 16$ | $0.134 \pm 0.020$ | $0.736 \pm 0.068$ |
| Gallbladder | 1.0% | $26 \pm 16$ | $0.153 \pm 0.047$ | $0.704 \pm 0.192$ |

**Consistent spatial distribution.** Notably, the spatial density of primitives (number per mm³, not shown in table) remains relatively uniform across organs (0.0015–0.0023 primitives/mm³), indicating that our method does not require higher primitive density for smaller organs. This uniform coverage suggests that the decoder has learned an efficient and scale-invariant representation strategy.

These findings validate that Gaussian splatting-based decoders learn anatomically meaningful representations. The automatic adaptation of primitive count, radius, and intensity to organ-specific characteristics demonstrates the flexibility of Gaussian parameterization for medical image reconstruction, without requiring architectural modifications or organ-specific hyperparameters.

