# OpenReview forum: "MedGMAE: Gaussian Masked Autoencoders for Medical Volumetric Representation Learning"
_ICLR.cc/2026/Conference — ICLR 2026 Poster_

### Official Review · Reviewer_fEUr · 2025-10-22

**Soundness:** 3
**Presentation:** 1
**Contribution:** 2
**Rating:** 2
**Confidence:** 4

**Summary:**

The authors propose a new representation learning framework based on learning 3D Gaussian primitives, which are then used to render volumes through Gaussian splatting. They argue that this intermediate representation enables the decoder to capture anatomical priors, improving performance on downstream tasks such as CT volume reconstruction, medical image classification, segmentation, and registration. The results are shown on multiple datasets and compared with relevant baselines.

**Strengths:**

Strengths of the paper:

1. The study includes a comprehensive evaluation across multiple medical downstream tasks, including CT volume reconstruction, image segmentation, disease classification, and deformable image registration, demonstrating the versatility and generalizability of the proposed representation learning framework.

2. To the best of our knowledge, the use of 3D Gaussian primitives for representation learning in medical volumetric data has not been explored before. This work introduces a novel perspective by leveraging Gaussian primitives as an intermediate representation to capture anatomical structure and spatial context, enabling more effective learning for a range of medical imaging tasks.

**Weaknesses:**

Weakness of the paper:

1. One major weakness of the paper is the lack of methodological novelty. The proposed approach is essentially identical to that of [1], with substantial portions(equations, query tokens, masking ratio) of the methodology directly adapted from it, the only difference being its application to 3D medical volumes instead of 2D images. Therefore, I assign a score of zero for the novelty of the proposed method.

(a) The reasons mentioned in the introduction line 048, 052 are exactly the same reasons for [1].

(b) Regarding the term “sparse representation,” it is unclear how the authors justify calling their representation sparse. Even in [1], the learned representation is not sparse by definition. It may be described as disentangled, but not sparse. The use of this term is confusing and should be clarified or corrected by the authors.


2. If i consider this a benchmark paper and not a methodology paper there are several things missing:

(a) No where in the paper i found how many Gaussian's are used for representation learning ? What is the impact of number of gaussians?

(b) What is the impact of different masking ratios ? As the authors mentioned in the paper that medical images have less textural changes compared to natural images, then why use the same masking ratio ? Feels illogical and uninformative.

(c) MAE are only compared in Table 1, missing from other tables.

In my opinion, as a benchmark, the work is not comprehensive and requires further development to provide more meaningful or complete evaluations.

(3) The code could have been released through an anonymized GitHub repository; however, this is a minor issue.


[1] Rajasegaran, Jathushan, Xinlei Chen, Rulilong Li, Christoph Feichtenhofer, Jitendra Malik, and Shiry Ginosar. "Gaussian masked autoencoders." arXiv preprint arXiv:2501.03229 (2025).

**Questions:**

The main question i have are :

1. Why do the authors refer to the proposed approach as a “sparse representation”? Please clarify what is meant by “sparse” in this context.

2. How does this method differ from “Gaussian Masked Autoencoders”? From my understanding, the approach appears to be largely identical, except for its application to 3D medical images. If there are substantive differences, the authors should provide a detailed comparison highlighting which components are new and which are retained from the original method.

3. Since the code for “Gaussian Masked Autoencoders” has not been publicly released, did the authors collaborate with the original developers or otherwise verify that their implementation and assumptions are consistent with the original work?

4. Please clarify the rationale behind using the same hyperparameters (e.g., masking ratio = 0.75) and other modeling choices. Were these parameters empirically validated for 3D medical data, or simply adopted from the original paper.

---

> ### Author Response · Authors · 2025-11-28
>
> We sincerely thank the reviewer for recognizing our  comprehensive evaluation, and the first application of 3D Gaussian primitives for representation learning to medical volumetric pretraining. Here we  provide detailed responses and additional experimental results.
>
> **Re Q1 & w1(b): "Sparse representation"**
>
> We acknowledge that our use of "sparse representation" may not align with the strict definition in machine learning literature, where sparsity typically refers to representations with many zero-valued elements. To clarify:
> What we mean by "sparse": Our Gaussian representation uses only 512 primitives (512 × 11 ≈ 5,632 parameters) to capture anatomically meaningful regions, compared to voxel-based methods that densely represent the entire volume (96³ ≈ 884,736 voxels). This constitutes a 99.25% parameter reduction.
>
> **Re Q2 & w1(a)：Novelty and Relationship with 2D GMAE**
>
> We acknowledge that both works share the high-level insight of using Gaussian primitives in masked autoencoders, but we respectfully argue that our contributions are substantively different and go beyond a simple 2D-to-3D extension.
>
> **Fundamental Differences in Problem Setting and Objectives**
>
> Our work was **not motivated by adapting 2D GMAE[1] to 3D**. Rather, we independently developed MedGMAE to address specific challenges in **3D medical volumetric representation learning**:
>
> - **2D GMAE[1] focuses on 2D natural images** with goals of learning layered 2.5D representations for zero-shot figure-ground segmentation and edge detection;
> - **MedGMAE targets 3D medical volumes** where the key challenges are: (i) modeling continuous anatomical structures across slices, (ii) efficient representation under sparse anatomical occupancy (~11.8%), and (iii) enabling zero-shot initialization for clinical CT reconstruction workflows
>
> These are fundamentally different research questions arising from different domains.
>
> **Technical Differences: What is Actually New**
>
> | Aspect | GMAE [1] | MedGMAE (Ours) |
> |--------|----------|----------------|
> | Gaussian Parameters | 14-D: {p, s, φ, r, o} (3D position, scale, rotation, RGB color, opacity) | 11-D: {μ, s, φ, I} (3D position, scale, rotation, single-channel intensity) |
> | Rendering Target | 2D pixel space via splatting | 3D volumetric space via Gaussian field aggregation (Eqs. 1-2) |
> | Loss Function | 2D pixel-space MSE | 3D masked-volume MSE |
> | Decoder Input | X_dec = {x̂ᵢ}∪{qⱼ} | X_dec = {x̂₁} ∪ {qⱼ} ∪ {x̂ᵢ}₂ⁿ (explicit class token + query tokens + visible tokens, Eq. 3) |
> | Parameter | Single parameterization for 14-D output | Four specialized heads (position/scale/rotation/intensity) with medical-specific activations |
> | Decoder Post-Training | Discarded  | Transferable: zero-shot initializer for 3DGS-based CT reconstruction |
> | Zero-Shot Capabilities | Figure-ground segmentation, layering, edge detection (2D) | CT reconstruction acceleration |
> | Code Availability | Not released | |
>
>
> **Re Q3**
> We did not collaborate with the original developers. Our team focuses on 3D medical imaging research and explores new possibilities to advance 3D medical tasks.
>
> **Re Q4 & w2: Hyperparameters Analysis**
> We provide comprehensive hyperparameters ablation studies  (you can find detailed results in Appendix Sec. Hyperparameter Sensitivity Analysis):
>
> Table R2: Impact of Number of Gaussians (K)
> | Mask Ratio | K | Recon L2 error | Segmentation (DSC%) |  | Zero-shot Inference for Reconstruction |  |  |  |  |
> |------------|---|-------------------|---------------------|--|---------------------|----------------------------------------|--|--|--|
> |  |  |  | AMOS 100% | SegTHOR 100% | Training Time (min) | Iter (PSNR>35) | Iter (SSIM>0.90) | PSNR(full) | SSIM(full) |
> | 25%        | 512     | 0.0031         | 80.03               | 86.03        | 541.5                                  | 1250           | 1050             | 44.6       | 98.4&plusmn;0.39     |
> | 50%        | 512     | 0.0068         | 79.01               | 86.00        | 572.4                                  | 1220           | 1040             | 45.3       | 98.5&plusmn;0.30     |
> | **75%**    | **512** | **0.0117**     | 84.90               | 89.15        | **357.0**                              | **1040**       | 980              | **46.2**   | **98.5**&plusmn;0.29 |
> | 85%        | 512     | 0.0147         | **85.17**           | **89.17**    | 545.9                                  | 1050           | 1000             | 46.0       | 98.3                 |
> | | | | | | | | | | |
> | 75% | 256 | 0.0143 | 79.99 | 85.82 | 498 | 1220 | 1080 | 45.85 | 98.43 |
> | 75% | **512** | **0.0117** | **84.90** | **89.15** | **357** | **1040** | **980** | **46.20** | **98.50** |
> | 75% | 768 | 0.0144 | 79.33 | 85.40 | 488 | 1140 | 1040 | 45.14 | 98.35 |
> | 75% | 1024 | 0.0137 | 78.87 | 85.89 | 546 | 1070 | 920 | 45.24 | 98.45 |
>
> **Re w3**: The anonymized GitHub repository is: https://anonymous.4open.science/r/MedGMAE-EC8F/index.html

---

### Official Review · Reviewer_Nwsv · 2025-10-26

**Soundness:** 3
**Presentation:** 3
**Contribution:** 3
**Rating:** 6
**Confidence:** 3

**Summary:**

This paper introduces MedGMAE (Medical Gaussian Masked Autoencoder), a novel self-supervised learning framework for volumetric medical image representation that replaces conventional voxel-based masked autoencoding with reconstruction via 3D Gaussian primitives. Traditional 3D masked autoencoders reconstruct missing voxel intensities directly, which the authors argue fails to capture continuous anatomical geometry, produces non-transferable decoders, and wastes parameters due to sparse anatomical occupancy. MedGMAE instead predicts a set of continuous 3D Gaussian parameters—position, scale, rotation, and intensity—to represent anatomical structures compactly and coherently. The decoder serves a dual purpose: it not only supports self-supervised pretraining but also provides a zero-shot, geometry-aware initialization for 3D Gaussian Splatting (3DGS)–based CT reconstruction. The framework achieves 99% parameter reduction relative to voxel-based MAEs and enables a 1.39× speed-up in CT reconstruction convergence. Experimental results on multiple medical imaging benchmarks—AMOS, FLARE’22, BTCV, SegTHOR (segmentation); CT-RATE (classification); IXI/OASIS (registration); and AAPM-Mayo (reconstruction)—show that MedGMAE consistently outperforms state-of-the-art voxel-based and self-supervised baselines (e.g., MAE, HySparK, VoCo, SUP) in both data-efficient and full-data regimes

**Strengths:**

- MedGMAE is the first to apply 3D Gaussian parameter prediction within a masked autoencoder framework for medical volumetric pretraining. This fundamentally shifts the pretraining objective from discrete voxel regression to continuous geometric modeling, aligning better with the smooth, anatomical nature of medical volumes.
- The model architecture (Figure 2, p. 3) elegantly integrates a ViT encoder with a lightweight Gaussian decoder and differentiable volumetric renderer. The authors detail parameterization (position, scale, rotation, intensity), initialization schemes, and differentiable rendering—showing a mature engineering of the 3D Gaussian field within a self-supervised context.
- Unlike typical masked autoencoders whose decoders are discarded post-pretraining, MedGMAE’s Gaussian decoder is explicitly transferable. It serves as a zero-shot initializer for downstream 3D Gaussian Splatting CT reconstruction, bridging pretraining and reconstruction in a unified formulation.

**Weaknesses:**

- While intuitive, the paper provides little formal reasoning or analysis explaining why Gaussian parameterization yields better anatomical representation. A discussion relating spatial frequency content or anatomical topology to Gaussian smoothness would strengthen the conceptual core.
- The custom CUDA Gaussian renderer adds substantial implementation complexity, and while Appendix §7.5 details optimizations, performance and stability under different resolutions or anisotropic voxel spacing are not benchmarked.
- The model produces geometric primitives, yet the paper stops short of analyzing their spatial or anatomical semantics

**Questions:**

- How does the number of Gaussian primitives (k) affect the trade-off between geometric fidelity and representation compactness? Have you explored adaptive Gaussian selection based on anatomical density?
- Cite RAPTOR which does some work on volumetric scans using foundation models (https://arxiv.org/abs/2507.08254)
- Can you provide quantitative evidence linking Gaussian coherence (e.g., overlap or smoothness metrics) to segmentation or registration performance?
- How much computational overhead does the Gaussian rendering add compared to standard voxel decoders, particularly during pretraining?
- Did you observe any degenerate Gaussian cases (e.g., extreme scales or vanishing densities), and how are these handled during training?
- Could the method support partially observed 3D data (sparse-view, partial slices) as a more direct input?
-

---

> ### Author Response · Authors · 2025-11-28
>
> We sincerely thank the reviewer for recognizing our novel architectural design, transferable decoder, and the first application of Gaussian-based SSL to medical volumetric pretraining. Here we answer the questions, and we are happy to answer more questions or run more experiments for the rebuttal.
>
> **Re Q1：Number of Gaussian primitives**
> We provide comprehensive ablation studies including the impact of Gaussian count (K). Detailed results are available in Appendix Sec. Hyperparameter Sensitivity Analysis：
>
> Table R1: Impact of Number of Gaussians (K)
> | Mask Ratio | K | Recon L2 error | Segmentation (DSC%) |  | Zero-shot Inference for Reconstruction |  |  |  |  |
> |------------|---|-------------------|---------------------|--|---------------------|----------------------------------------|--|--|--|
> | 75% | 256 | 0.0143 | 79.99 | 85.82 | 498 | 1220 | 1080 | 45.85 | 98.43 |
> | 75% | **512** | **0.0117** | **84.90** | **89.15** | **357** | **1040** | **980** | **46.20** | **98.50** |
> | 75% | 768 | 0.0144 | 79.33 | 85.40 | 488 | 1140 | 1040 | 45.14 | 98.35 |
> | 75% | 1024 | 0.0137 | 78.87 | 85.89 | 546 | 1070 | 920 | 45.24 | 98.45 |
>
> In our ablation experiments, K=512 Gaussian primitives provides the optimal efficiency-performance trade-off, avoiding both under-parameterization (insufficient representation capacity) and over-parameterization (unnecessary computational overhead).
>
> **Adaptive Gaussian selection based on anatomical density:**
> While our current implementation uses a fixed K=512, future extensions could employ gated Mixture-of-Experts to adaptively allocate Gaussians based on anatomical complexity, achieving better representation with minimal computational overhead.
>
> **Re Q2:**
> Thank you for the valuable suggestion. We have incorporated the RAPTOR citation in the revised manuscript. Specifically, we added the following discussion at the end of the Introduction section:
> "Recent work explores train-free paradigms that leverage pretrained 2D foundation models to extract semantic information from 3D volumes \citep{2025raptor}, demonstrating an alternative approach to volumetric representation learning."
>
> **Re Q3 & Weakness 1：Formal reasoning**
>
> Our volumetric field is a continuous mixture of anisotropic Gaussians:
> $$V(\mathbf{X}) = \sum_{i} I_i \exp \left( -\frac{1}{2} (\mathbf{X} - \mu_i)^{\text{T}} \Sigma_i^{-1} (\mathbf{X} - \mu_i) \right)$$
>
> The Fourier transform reveals the spectral structure:
> $$\mathcal{F}\{V\}(\mathbf{k}) = \sum_{i} I_i (2\pi)^{3/2} |\Sigma_i|^{1/2} \exp\left(-i \mathbf{k} \cdot \mu_i - \frac{1}{2} \mathbf{k}^{\text{T}} \Sigma_i \mathbf{k}\right)$$
>
> Key insight: Each Gaussian acts as an **anisotropic low-pass filter** where $\mathbf{k}^T\Sigma_i\mathbf{k}$ induces direction-dependent attenuation—suppressing high-frequency noise perpendicular to organ boundaries while preserving edge orientations along anatomical axes.
>
> Contrast: Voxel-based MAE performs independent per-location regression without explicit frequency constraints, resulting in spectral content determined by network capacity rather than anatomical structure.
>
> Empirical validation (appendix): Power spectrum analysis shows Gaussian MAE exhibits significantly lower variance in high-frequency regime with smoother spectral decay. Organ-specific analysis reveals automatic primitive allocation proportional to anatomy—larger organs receive more Gaussians, radii adapt to geometric complexity. See updated appendix for details.
>
> **Re Q4: Computational overhead**
> We provide detailed profiling results (batch=1, 3090 GPU, 100 iters):
> Decoder-only overhead (pretraining relevant):
> |      | Voxel MAE | MedGMAE | Diff   |
> |-|-|-|-|
> | Time (ms) | 5.92  | 5.62    | -0.3  |
> | Memory(MB)| 93.48     | 147.66  | +55 |
> | FLOPs(GFs)| 9.236     | 8.898   | -0.3  |
>
> The Gaussian decoder is faster than voxel decoders because it outputs only 11 parameters per Gaussian versus 1,728 voxels (12³), reducing output layer FLOPs by 99.4% (2.2M vs 340M).
> **With rendering (only pretraining loss computation):**
> - Sparse rendering (masked regions only):  147ms/patch (save 75% vs full)
> - Memory remains (full or sparse rendering): 147.66MB
>   The modest pretraining cost is offset by superior transfer performance and zero-shot CT reconstruction capabilities.
>
> **Re Q5: Degenerate Gaussian cases**
> Yes, we mitigate degenerate cases through customized initialization: scale biases (-1.386 → ~0.2 after sigmoid) prevent extreme scales, and density biases (-0.405 → ~0.5) avoid vanishing densities. This ensures stable training (details in Appendix Sec. 7.4).
>
> **Re Q6:**
> Current pretraining uses random masking on complete volumes. For partially observed data,
> Partial slices: Feasible through zero-padding of missing slices
> Sparse 2D views: Requires architecture modification (current encoder expects volumetric input)
>
> The latter represents a direction: **learning to infer 3D Gaussian representations directly from sparse 2D projections** would enable applications in low-dose CT.

---

### Official Review · Reviewer_w2RZ · 2025-10-30

**Soundness:** 3
**Presentation:** 4
**Contribution:** 3
**Rating:** 8
**Confidence:** 4

**Summary:**

The paper introduces a method for self-supervised pretraining for medical volumes, based on 3D Gaussian primitives (instead of traditional volumetric representations). The idea is that the Gaussian primitives can capture long-range dependencies better than local voxel-based representations. Results are given for multiple datasets and tasks.

**Strengths:**

- Good clear paper, explaining the context and method well.

- The motivation for the method seems reasonable.

- Extensive experiments are included for various datasets and tasks.

- Computational requirements are explicitly discussed.

**Weaknesses:**

- The authors use 96x96x96 voxel cubes, which is relatively low-resolution. It remains unclear to me how the method would perform on higher resolution data. For example, medical CT data can be 512^3 up to 2048^3 (see e.g. [1]). Non-medical CT datasets can be even larger. One concern is whether, for higher resolutions, the 3D gaussians are still able to capture long-range correlations well, and required computation time does not become prohibitively large.

- It is not clear how specific hyperparameters settings are chosen by the authors, and how results are affected by different choices for the hyperparameters. This is also try for comparison methods -- how are the hyperparameters chosen for these?

- Results are not always presented clearly -- for example, in Table 3 the caption states that the best result is shown in bold, but TransMorph achieves the best Dice result but isn't bold.

[1] Oostveen, L. J., Boedeker, K. L., Brink, M., Prokop, M., de Lange, F., & Sechopoulos, I. (2020). Physical evaluation of an ultra-high-resolution CT scanner. European radiology, 30(5), 2552-2560.

**Questions:**

- How does the method perform (both in terms of accuracy, and computation time) for larger volumes that can be common in practice?

- How were hyperparameters chosen, and how do hyperparameters affect results?

---

> ### Author Response · Authors · 2025-11-28
>
> We sincerely thank the reviewer for recognizing our clear presentation, reasonable motivation, and extensive experimental validation. Here we  provide detailed responses and additional experimental results.
>
> **Response Weakness 1 & Q1 :  Scalability to Higher Resolution Data**
>
> We appreciate this important practical concern. We would like to clarify how our method handles resolution scaling:
>
> Patch-based Processing Design:
> Due to memory constraints, medical imaging models process volumes via sliding windows—we use 96³ patches as input units. Our 512 Gaussians represent each patch, meaning the maximum correlation range is bounded by the patch size, not the full volume resolution.
>
> Resolution Scaling Behavior:
> For higher-resolution volumes (e.g., 512³ or 2048³):
> - Representation capacity remains constant: Each 96³ patch still uses 512 Gaussians
> - Long-range correlation: Maintained within each patch; cross-patch dependencies handled by sliding window inference
> - Computation time scales linearly with number of patches (e.g., 512³ volume = ~152 patches vs. ~1 patch for 96³)
> - Gaussian efficiency: Overly large or small Gaussians scale would reduce rendering computational efficiency
>
> **Key Insight:**
> The Gaussian primitives capture local anatomical structures within patches. Higher resolutions increase inference time proportionally to patch count, but per-patch representation quality and efficiency remain unchanged.
>
> **Response Weakness 2 & Q2:   Hyperparameter Sensitivity (K-Gaussian Number & Mask Ratio)**
> We provide comprehensive ablation studies addressing both concerns with systematic hyperparameter sweeps  (you can find detailed results in Appendix Sec. Hyperparameter Sensitivity Analysis):
>
> Table R1: Impact of Number of Gaussians (K)
> | Mask Ratio | K | Recon L2 error | Segmentation (DSC%) |  | Zero-shot Inference for Reconstruction |  |  |  |  |
> |------------|---|-------------------|---------------------|--|---------------------|----------------------------------------|--|--|--|
> |  |  |  | AMOS 100% | SegTHOR 100% | Training Time (min) | Iter (PSNR>35) | Iter (SSIM>0.90) | PSNR(full) | SSIM(full) |
> | 25%        | 512     | 0.0031         | 80.03               | 86.03        | 541.5                                  | 1250           | 1050             | 44.6       | 98.4&plusmn;0.39     |
> | 50%        | 512     | 0.0068         | 79.01               | 86.00        | 572.4                                  | 1220           | 1040             | 45.3       | 98.5&plusmn;0.30     |
> | **75%**    | **512** | **0.0117**     | 84.90               | 89.15        | **357.0**                              | **1040**       | 980              | **46.2**   | **98.5**&plusmn;0.29 |
> | 85%        | 512     | 0.0147         | **85.17**           | **89.17**    | 545.9                                  | 1050           | 1000             | 46.0       | 98.3                 |
> | | | | | | | | | | |
> | 75% | 256 | 0.0143 | 79.99 | 85.82 | 498 | 1220 | 1080 | 45.85 | 98.43 |
> | 75% | **512** | **0.0117** | **84.90** | **89.15** | **357** | **1040** | **980** | **46.20** | **98.50** |
> | 75% | 768 | 0.0144 | 79.33 | 85.40 | 488 | 1140 | 1040 | 45.14 | 98.35 |
> | 75% | 1024 | 0.0137 | 78.87 | 85.89 | 546 | 1070 | 920 | 45.24 | 98.45 |
>
> These comprehensive ablation studies validate our design choices and demonstrate the robustness of MedGMAE across different configurations for the datasets that we used during experiments.(1) The 75\% mask ratio consistently emerges as near optimal for both transfer learning and zero-shot reconstruction, balancing reconstruction quality with learned representation quality; (2) "K=512 Gaussian primitives" provides the best efficiency-performance trade-off, avoiding both under-parameterization and over-parameterization; (3) Our method's performance remains stable within reasonable hyperparameter ranges.
>
> **Response Weakness 3:**
> Thank you for catching this formatting question. We have corrected the bold formatting in Table 3.

---

### Official Review · Reviewer_TUT3 · 2025-10-31

**Soundness:** 2
**Presentation:** 2
**Contribution:** 2
**Rating:** 6
**Confidence:** 3

**Summary:**

This paper proposes MedGMAE, a self-supervised pre-training framework for 3D medical images that replaces voxel-level masked reconstruction with prediction of sets of 3D Gaussian primitives followed by differentiable volumetric rendering. The encoder is ViT-based; a lightweight decoder emits k Gaussians which are rendered and compared to masked ground truth. An extended variant (MedGMAE\*) adds hierarchical residual blocks for coarse-to-fine Gaussian refinement. The authors claim that Gaussian reconstruction better matches anatomical continuity, yields a transferable decoder that can zero-shot initialize 3D Gaussian–based CT reconstruction, and is parameter efficient versus voxel MIM. Experiments show gains on segmentation, classification, registration, and faster convergence for low-dose CT reconstruction when using the decoder as an initializer.

**Strengths:**

1. The paper precisely formulates predicting Gaussian sets, detailing tokenization, query tokens, parameter heads, activations/normalization, and rendering, including initialization choices for stability.
2. The motivation is well-argued, and the limitations of voxel-level objectives for continuous anatomy and the sparsity/efficiency argument for Gaussians are articulated, with a schematic overview (Fig. 1) that connects sparsity to parameter reduction and speedup.
3. On segmentation with 1% labels, MedGMAE outperforms prior best notably on AMOS and FLARE’22 ; classification and registration show competitive/better results as well.  The ablation studies further support the claim.

**Weaknesses:**

1. The related work briefly contrasts with 2D Gaussian MAE (Rajasegaran et al., 2025) and several medical 3DGS papers, but head-to-head empirical comparisons versus the most relevant medical Gaussian pretext tasks (if any) or Gaussian-rendered proxy tasks are missing; the comparison set is largely voxel-MIM or contrastive SSL. This makes it hard to attribute the transfer gains specifically to Gaussian reconstruction rather than dataset/implementation choices.
2. The paper states UNETR fine-tuning with consistent preprocessing, but various baselines load official pre-trained weights. Different pre-training corpora/compute may advantage some methods. I suggest the authors to add an explicit data-scale or compute-matched comparison.
3. The method fixes yet there is no systematic study of k on transfer and reconstruction. The claim of 99.25% parameter reduction relies on a specific setting; a sensitivity sweep would help validate efficiency-generalization tradeoffs.
4. The main text lists very large pretraining settings (e.g., “batch size 192, 400K steps” vs. Appendix Table 9 “batch size 8, max steps 100K”), making it unclear which configuration corresponds to the reported results; this needs consolidation.

**Questions:**

Please refer to the weaknesses part, especially weakness 2&3.

---

> ### Author Response · Authors · 2025-11-28
>
> We sincerely thank the reviewer for the positive comments on our work's "precise formulation of Gaussian sets prediction," "well-argued motivation and articulated limitations of voxel-level objectives," and "notable performance improvements on segmentation tasks". Here we address the concerns and questions raised, and we are happy to provide detailed responses and additional experimental results.
>
> **Response Weakness 1: Lack of Gaussian-Based Baseline Comparisons**
>
> Why Direct Comparisons Are Infeasible?
>
> **2D Gaussian MAE** (Rajasegaran et al., 2025): Code unavailable; their 2D design (using z-axis for abstract 2.5D layers) is fundamentally incompatible with true 3D volumetric anatomy.
> **Medical 3DGS papers**: All existing works are task-specific supervised reconstruction methods, not SSL pretrain task.
> **MedGMAE** is the first Gaussian-based SSL method for medical imaging domain.
>
> **Response Weakness 2: Data-Scale and Compute Matching**
>
> We present a comprehensive comparison of pretraining data across all methods in Table R1. Importantly, MedGMAE uses significantly less pretraining data (about 5k volumes) compared to most baselines (about 10k volumes), yet achieves superior downstream performance.
>
> Table R1: Pretraining Data Scale Comparison
> | Method | Pretraining Datasets | Volume Count |
> |--------|---------------------|--------------|
> | MAE | LUNA16, COVID-19, BTCV, Sliver07, CT-ORG, FLARE'22, CHAOS, NaH-Seg, KiPA22, Pancreas-CT, LiTS, AbdomenCT-1k, AbdomenAtlasMini1.0 | ~10,000 |
> | VoCo | TCIA Covid19, Luna16-jx, BTCV, MM-WHS, Spleen, STOIC2021, Totalsegmentator, Flare23, LiDC, HNSCC | ~10,000 |
> | HySparK | BTCV, CHAOS, WORD, FLARE'22, AbdomenCT-1k, TotalSegmentator, AMOS22 | ~6,814 |
> | MedGMAE | AbdomenAtlas 1.0Mini | ~5,195 |
>
> *AbdomenAtlas 1.0Mini aggregates: Pancreas-CT, LiTS, KiTS, AbdomenCT-1k, CT-ORG, MSD CT, BTCV, AMOS22, WORD, FLARE'23, AbdomenCT-12organ. Data overlap: AbdomenAtlas 1.0Mini focuses on abdominal CT volumes, which are **fully subsumed** within the larger pretraining corpora of MAE/VoCo/HySparK.
>
> With the exception of VoCo (batch size 4), MAE, HySparK, and MedGMAE employed a batch size of 8 for ~100k steps during pretraining.
>
> These baselines include our entire data domain (abdominal CT) plus additional breast and brain CT. Despite using a **subset** of competitors' data domains, MedGMAE achieves competitive performance, demonstrating that gains stem from representation learning effectiveness rather than data advantage.
>
> **Response weakness 4:**
> Corrected Configuration: We acknowledge an text question in Section 4.2. The revision now correctly states: 'RandCropByPosNegLabeld (3 crops/volume), batch size 8, 100k steps,' consistent with Appendix Table 9."
>
> **Response Weakness 3: Hyperparameter Sensitivity (K-Gaussian Number & Mask Ratio)**
>
> We provide comprehensive ablation studies addressing both concerns with systematic hyperparameter sweeps  (you can find detailed results in Appendix Sec. Hyperparameter Sensitivity Analysis):
>
> Table R2: Impact of Number of Gaussians (K)
> | Mask Ratio | K | Recon L2 error | Segmentation (DSC%) |  | Zero-shot Inference for Reconstruction |  |  |  |  |
> |------------|---|-------------------|---------------------|--|---------------------|----------------------------------------|--|--|--|
> |  |  |  | AMOS 100% | SegTHOR 100% | Training Time (min) | Iter (PSNR>35) | Iter (SSIM>0.90) | PSNR(full) | SSIM(full) |
> | 25%        | 512     | 0.0031         | 80.03               | 86.03        | 541.5                                  | 1250           | 1050             | 44.6       | 98.4&plusmn;0.39     |
> | 50%        | 512     | 0.0068         | 79.01               | 86.00        | 572.4                                  | 1220           | 1040             | 45.3       | 98.5&plusmn;0.30     |
> | **75%**    | **512** | **0.0117**     | 84.90               | 89.15        | **357.0**                              | **1040**       | 980              | **46.2**   | **98.5**&plusmn;0.29 |
> | 85%        | 512     | 0.0147         | **85.17**           | **89.17**    | 545.9                                  | 1050           | 1000             | 46.0       | 98.3                 |
> | | | | | | | | | | |
> | 75% | 256 | 0.0143 | 79.99 | 85.82 | 498 | 1220 | 1080 | 45.85 | 98.43 |
> | 75% | **512** | **0.0117** | **84.90** | **89.15** | **357** | **1040** | **980** | **46.20** | **98.50** |
> | 75% | 768 | 0.0144 | 79.33 | 85.40 | 488 | 1140 | 1040 | 45.14 | 98.35 |
> | 75% | 1024 | 0.0137 | 78.87 | 85.89 | 546 | 1070 | 920 | 45.24 | 98.45 |
>
> These comprehensive ablation studies validate our design choices and demonstrate the robustness of MedGMAE across different configurations for the datasets that we used during experiments. "K=512 Gaussian primitives" provides the best efficiency-performance trade-off, avoiding both under-parameterization and over-parameterization;

---

### Comment · Area_Chair_LaZs · 2025-11-25
**Please engage with the reviewers during the rebuttal/discussion phase (if the authors plan to)**

Dear authors,

It has been over one week since the review comments were released.

If you plan to join the rebuttal, please engage sooner rather than later, so that the reviewers can have time engage as well.

Thanks,
AC

---

### Author Response · Authors · 2025-12-01
**Summary for Area Chair**

## Paper Overview

MedGMAE addresses fundamental limitations in medical volumetric self-supervised learning by replacing discrete voxel-level reconstruction with 3D Gaussian primitive prediction. Our key insight is that medical anatomy—characterized by sparse occupancy and continuous geometric structures—is better captured through compact Gaussian representations than dense voxel grids.

## Consensus Strengths

- Methodological contributions: First Gaussian-based SSL framework for medical volumetric pretraining (Reviewers TUT3, w2RZ, Nwsv, fEUr)
- Technical rigor: Precise formulation of Gaussian prediction, well-engineered architecture with specialized parameter heads and initialization schemes (Reviewers TUT3, Nwsv)
- Comprehensive evaluation: Strong empirical results across 7 datasets and 4 task types, demonstrating versatility and translational potential (Reviewers w2RZ, Nwsv, fEUr)
- Clear presentation: Well-motivated problem formulation and articulated limitations of voxel-based objectives (Reviewers TUT3, w2RZ)

## Detailed Response

### 1. Hyperparameter Sensitivity Analysis (All Reviewers)

Suggestion: Systematic evaluation of pretraining Gaussian number (K) and mask ratio

**Re**: Comprehensive ablation studies added to **Appendix: Hyperparameter Sensitivity Analysis**, sweeping K ∈ {256, 512, 768, 1024} and mask ratio ∈ {25%, 50%, 75%, 85%}. Results validate K=512 achieves optimal efficiency-performance tradeoff; 75% mask ratio balances reconstruction quality and representation learning (85% shows some improvement in segmentation but significantly worse reconstruction efficiency).

---

### 2. Methodological Novelty vs 2D GMAE (fEUr)

Question: Differentiation from Rajasegaran et al.'s 2D Gaussian MAE [2025].

**Re**: While the Related Work section established conceptual distinctions, we now provide detailed technical comparison demonstrating fundamental differences:

**Problem setting**: 2D natural images (abstract 2.5D layering) vs 3D medical volumes (continuous anatomical structures, sparse occupancy, ct reconstruction)

**Technical architecture**:
- Gaussian parameterization: 14-D RGB+opacity vs 11-D single-channel intensity
- Rendering: 2D pixel splatting vs 3D volumetric field aggregation
- Decoder design: Single parameterization vs four specialized medical-specific heads with custom initialization
- Post-training utility: Discarded vs transferable zero-shot CT reconstruction initializer

Not domain transfer—fundamentally different research questions arising from medical imaging challenges.

---

### 3. Dataset Fairness (TUT3)

Question: Pretraining data/compute advantages may confound comparisons.

**Re**: Comprehensive data comparison provided. Despite using less data, MedGMAE achieves superior performance, demonstrating gains stem from representation learning effectiveness, not data advantage.

---

### 4. Computational Overhead & Scalability (w2RZ, Nwsv)

Question:
- Scalability to larger resolutions (512³–2048³)
- CUDA rendering computational cost

**Re**:

Scalability: Patch-based design (96³ patches + sliding window inference) scales linearly with volume size. Per-patch representation quality and efficiency remain constant—higher resolutions simply process more patches.

Overhead profiling (batch=1, 3090 GPU): Gaussian decoder is 5.1% faster than voxel decoder (fewer FLOPs: 8.898 vs 9.236 GF) with modest GPU memory overhead.

---

### 5. Theoretical Analysis (Nwsv)

Suggestion: Add formal reasoning why Gaussians improve anatomical representation.

**Re**: Added frequency-domain analysis to Appendix. Fourier transform reveals each Gaussian acts as an orientation-selective low-pass filter, suppressing high-frequency noise perpendicular to organ boundaries while preserving edge orientations along anatomical axes. Radial power spectrum analysis shows Gaussian MAE exhibits superior high-frequency coherence and smoother spectral decay. Organ-specific statistics demonstrate automatic primitive allocation proportional to anatomical complexity.

---

### 6. Minor Issues

Addressed comprehensively:
- Table 3 formatting corrected (w2RZ)
- Configuration unified to "batch size 8, 100k steps" (TUT3)
- RAPTOR citation added to Related Work ( Nwsv)
- Anonymous code released: https://anonymous.4open.science/r/MedGMAE-EC8F/ (fEUr)

---

## Conclusion

We have comprehensively addressed all reviewer comments. The rebuttal substantively strengthens the manuscript through:
- Systematic ablations validating all design choices across hyperparameters
- Detailed differentiation establishing fundamental novelty beyond 2D natural image methods
- Fairness comparison demonstrating efficiency with significantly less pretraining data
- Reasoning analysis providing frequency-domain explaination
- Practical validation confirming computational efficiency

The revised manuscript includes substantial additions to the Appendix, addressing all technical questions while maintaining concise main text presentation.

---

### Meta-Review · Area_Chair_EL1K · 2025-12-23

**Summary:**

Among the four reviewers, three (Nwsv, w2RZ, and TUT3) provided overall positive evaluations, and their related concerns were largely clarified in the authors’ response. Reviewer fEUr raised major concerns regarding the methodological novelty and pointed out the absence of certain hyperparameter analyses. The authors provided clarifications and supplemented their rebuttal with detailed experimental results, which the AC considers to have largely addressed this reviewer’s concerns. Overall, the work explores the potential of GMAE in the domain of 3D medical imaging, and makes a clear contribution to the field. Therefore, the AC recommends acceptance of the manuscript, with the expectation that the authors fully address and clarify all reviewers’ concerns in the final version.

**Reviewer Concerns:**

The four reviewers raised concerns in aspects such as hyperparameter sensitivity analysis, methodological novelty, dataset fairness, computational overhead, scalability, and theoretical analysis. The authors have, for the most part, addressed these concerns through their rebuttal, and there appear to be no fundamental flaws in the paper.

**Reviewer Scores:**

Given the detailed and persuasive rebuttal provided by the authors, the AC believes that reviewers Nwsv, w2RZ, and TUT3 are likely to maintain their positive evaluations, and reviewer fEUr may consider giving a positive score as well.

---

### Decision · Program_Chairs · 2026-01-26

Accept (Poster)